# EXPLORING NON-LINEARITY IN ATTENTION

## ABSTRACT

The representational ability of Transformer architectures arises from two sources of non-linearity: position-wise non-linearity via feed-forward layers and contextual non-linearity through self-attention. In this work, we revisit this distinction and pose two key questions: Can self-attention itself realize position-wise non-linearity? And is contextual non-linearity truly necessary? First, we prove that by appending a fixed bias vector into the input, stacked self-attention layers can approximate deep feed-forward networks—showing that attention alone is sufficient to implement position-wise non-linearity. Second, we prove that contextual non-linearity, i.e., input-dependent attention patterns, is not indispensable: fixed or even randomly chosen patterns, when combined with a feed-forward layer, can still produce context-sensitive representations of the same token in different contexts. As an application, we prove that a two-layer attention-only Transformer can accurately predict masked tokens in masked language modeling. Both theoretical analysis and empirical studies on pre-trained models and synthetic data support our theory.

## 1 INTRODUCTION

Non-linearity is a fundamental source of expressive power in modern deep learning architectures. At the core of the Transformer model (Vaswani et al., 2017) is the interactive composition of feed-forward layers and self-attention layers, which together give rise to two distinct forms of non-linearity. Contextual non-linearity emerges from the self-attention mechanism: through the combined effects of the dot product and the Softmax function, each self-attention layer induces a non-linear mapping from the input sequence to the attention weights. This non-linearity does not operate independently on each token, but instead arises from interactions among tokens within the same sequence. Such a mechanism enables the model to capture long-range dependencies, a capability widely regarded as one of the primary reasons why Transformers surpass recurrent and convolutional neural networks on a variety of sequence modeling tasks (Sherstinsky, 2020; O'shea & Nash, 2015).

Self-attention, however, is not the sole contributor to the expressivity of Transformer models. Each self-attention layer is followed by a position-wise feed-forward layer, which accounts for roughly two-thirds of the model's parameters. While the self-attention module computes a weighted linear aggregation of linearly transformed token representations, with weights determined by pairwise token similarity, the feed-forward network applies a non-linear transformation independently to each token embedding. This position-wise non-linearity complements the contextual non-linearity of attention, and together they form the core expressive machinery of the Transformer. In this work, we rethink self-attention mechanism through the lens of non-linear capacity. In particular, we aim to address the following two central questions:

**Is position-wise non-linearity exclusively introduced by feed-forward layers, or do certain attention heads also implicitly contribute to it?** Self-attention is designed to compute the interaction between tokens in different positions while feed-forward neural networks can be viewed as modeling interactions between tokens and a set of fixed tokens that encode global, input-independent information. Based on this similarity, a natural question is whether self-attention layers alone can behave like feed-forward neural networks, even deep ones. Prior works (Sukhbaatar et al., 2019; Huben & Morris, 2023) have investigated this problem from an empirical and theoretical perspective, respectively. As shown in Figure 2, we identify some attention heads in real-world language models that perform similar behavior to that of a feed-forward layer, meaning some attention heads implicitly implement position-wise non-linearity.

**Is contextual non-linearity indispensable, or can fixed and even random attention patterns still yield effective performance in certain tasks.** Unlike feed-forward neural networks, which apply exactly the same operations to each token, self-attention layers can incorporate contextual information. In the standard attention mechanism, the attention pattern is instance-dependent and vary across inputs. However, empirical results (Raganato et al., 2020; Tay et al., 2021; Kovaleva et al., 2019) and also Figure 1 have shown that the attention patterns learned by Transformers are often limited in diversity and can be replaced by instance-agnostic patterns without a decline in performance. These empirical results motivates us to theoretically study the effectiveness of input-independent attention mechanism.

Our contributions to answer the above two questions are summarized as follows:

- In Theorem 4.1, we extend the results in (Huben & Morris, 2023) to a more general setting by proving that self-attention can approximate deep feed-forward neural networks. In Appendix D, we provide empirical verification for Theorem 4.1. In addition, we identify that in Figure 2, there are some attention heads in **Bert** (Devlin et al., 2019) mostly computing the interaction between data tokens and special tokens [CLS] or [SEP], which perform similar behavior to that of a feed-forward neural network, meaning that some attention heads in real-world models implicitly provide position-wise non-linearity by letting the data tokens interacting with some special tokens that are global and independent of input sequences.

- In masked language modeling, Transformers are required to predict the original tokens from the token [MASK] based on the contexts it appears. As shown in Figure 3, the representation of [MASK] tokens are influenced by different contexts. To verify the effectiveness of instance-agnostic attention patterns, we theoretically prove that both a fixed attention pattern and an arbitrary attention pattern equipped with a feed-forward layer can distinguish the same tokens in different contexts. Specifically, we show that in Proposition 5.2 and 5.3, the same token [MASK] in different sequences can be mapped to different values by self-attention with input-independent attention patterns.

- As an application, we consider masked language modeling and prove that a two-layer attention-only Transformer is sufficient to make accurate predictions for masked tokens, where one layer implements position-wise non-linearity and the other aggregates contextual information (see Theorem 6.1).

- Observations on pre-trained language models (see Figure 1, 2, 3) and experiments on synthetic datasets (see Appendix D) support our theory.

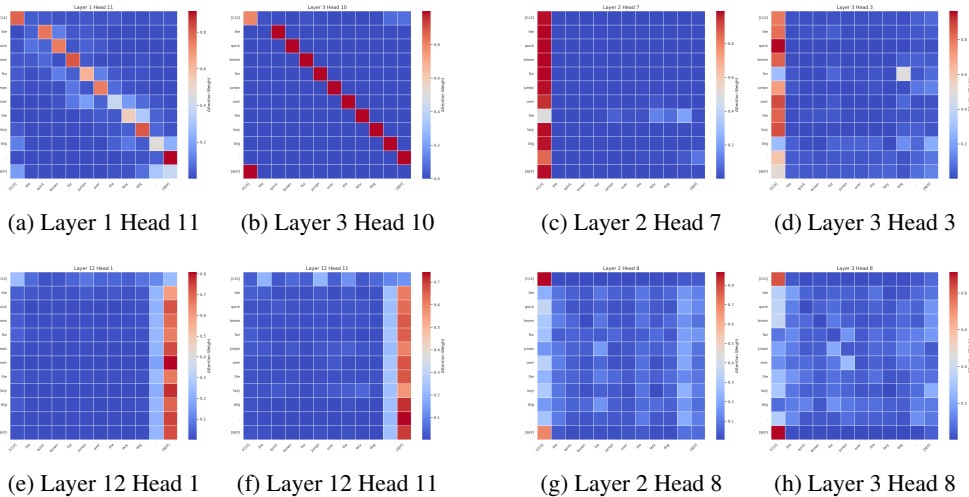

| (a) Layer 1 Head 11 | (b) Layer 3 Head 10 | (c) Layer 2 Head 7 | (d) Layer 3 Head 3 |

| (e) Layer 12 Head 1 | (f) Layer 12 Head 11 | (g) Layer 2 Head 8 | (h) Layer 3 Head 8 |

Figure 1: **Repeated Attention Patterns.** We identify several attention patterns repeatedly occurring in different attention heads in **Bert** (Devlin et al., 2019) (12 layers with 12 heads in each layer). (a) and (b) shows that each token attends to its next one. In (c) and (d), each token mostly interacts with the first token, that is, [CLS], while in (e) and (f), each token mainly interacts with the last token [SEP]. Similar observation can be found in (Kovaleva et al., 2019; Clark et al., 2019).

## 2 RELATED WORK

**Understanding Attention Mechanism.** Tsai et al. (2019) showed that the attention mechanism can be reformulated as a kernel smoother over inputs, where the kernel scores correspond to token similarity. This perspective provides a more principled understanding of the individual components of Transformer's attention mechanism, while also enabling competitive performance with deduced computation cost. Building on this view, a number of kernel-based variants have been developed, such as Performer (Choromanski et al., 2020), Skyformer (Chen et al., 2021), Linear Attention (Katharopoulos et al., 2020) and Sumformer (Alberti et al., 2023). Another line of research focuses on introducing sparsity into attention pattern to improve efficiency. Longformer, Beltagy et al. (2020), enables scalable processing of long documents without truncating the input. Zaheer et al. (2020) proposed BiGBIRD, a sparse attention mechanism that is linear in the number of tokens. Theoretically, they proved that BIGBIRD is a universal approximator and Turing complete. Similarly, Zhang et al. (2021) introduced a Softmax-free model, which avoids the categorical distribution and all negative attention scores are pruned out because of the use of ReLU activation function. Tay et al. (2021) proposed SYNTHESIZER, a model that learns synthetic attention weights directly, bypassing the computation of token-token interactions. Beyond the modification of attention mechanism, interpretability studies have investigated the representational properties of attention. Reif et al. (2019) explored the syntactic information encoded in attention matrices in BERT, while (Kovaleva et al., 2019) found that a limited set of recurring attention patterns emerge across different heads, and that manually disabling certain heads can even lead to performance improvement. In addition, Zhao et al. (2021) introduced a method to measure the degree of non-linearity of different components of Transformers. Concurrently, Dong et al. (2025) showed that Transformers with random attention patterns have a stable forward pass, which converges to a stochastic differential equation.

**Understanding Feed-Forward Layers.** The role of feed-forward layers have been closely examined. Geva et al. (2020) showed that feed-forward layers operate as Key-value memories, where each key correlates to textual patterns in the training example. Sukhbaatar et al. (2019) proposed a Transformer variant solely consisting of attention layers, in which the attention mechanism is augmented with persistent memory vectors that are able to substitute for feed-forward layers. Extending this, Huben & Morris (2023) theoretically proved that one attention layer can approximate one feed-forward layer with an extra bias vector appended to the input and special mask strategy. Xu et al. (2024) argued that the primary role of feed-forward layers is to provide non-linearity and they presented an improved FFN module, which is able to enrich non-linear signals while effectively reducing the hidden dimensionality. Kobayashi et al. (2023) showed that FFNs modify the input contextualization to emphasize specific types of linguistic compositions. Ikeda et al. (2025) investigated the layerwise importance of feed-forward layers in Transformer-based language models during pretraining, finding that concentrating FFNs in $70\%$ of the consecutive middle layers consistently outperforms standard configurations for multiple downstream downstream tasks. From another perspective, Bozic et al. (2023) analyzed the effectiveness of using standard shallow feed-forward neural networks to mimic the attention behavior. Zhang et al. (2020) empirically observed that replacing the upper self-attention layers in the encoder with feed-forward layers causes no performance degradation, and sometimes even yields minor gains. Finally, Liu et al. (2021) proposed a simple architecture, based on MLPs with gating, verifying that it can match Transformer's performance across a variety of tasks.

## 3 PRELIMINARIES

### 3.1 TRANSFORMER

The Transformer architecture is a mapping that takes a sequence $\boldsymbol{X} \in \mathbb{R}^{D \times N}$ composed of $n$ tokens each with an embedding size of $d$ as an input and outputs another sequence. In a Transformer neural network, there are two primary layers: a self-attention layer and a feed-forward layer. The attention layer mixes information across different positions in the sequence through dot products of token embeddings. Notably, Transformers utilize the Softmax function to transform each dot product into a probability or a weight, and the output of the attention layer becomes a linear combination of token embeddings according to these weights. Subsequently, the token-wise feed-forward layer operates independently on separate tokens without inter-token interaction.

A Transformer neural network is composed of alternating feed-forward layers and self-attention layers. An self-attention layer $\mathcal{F}_{SA} : \mathbb{R}^{D \times N} \to \mathbb{R}^{D \times N}$ with $H$ heads is defined as

$$\mathcal{F}_{SA}(\boldsymbol{X}) := \boldsymbol{X} + \sum_{i=1}^{H} \boldsymbol{W}_O^{(i)} \boldsymbol{W}_V^{(i)} \boldsymbol{X} \sigma_S \left[ \left( \boldsymbol{W}_K^{(i)} \boldsymbol{X} \right)^{\top} \left( \boldsymbol{W}_Q^{(i)} \boldsymbol{X} \right) + \boldsymbol{R}^{(i)} \right],$$

where $\boldsymbol{W}_K^{(i)}, \boldsymbol{W}_Q^{(i)}, \boldsymbol{W}_V^{(i)} \in \mathbb{R}^{S \times D}$ and $\boldsymbol{W}_Q^{(i)} \in \mathbb{R}^{D \times S}$ are the weight matrices, $\boldsymbol{R}^{(i)} \in \mathbb{R}^{N \times N}$ is the mask matrix, which satisfies $\boldsymbol{R}_{i,j}^{(i)} = -\infty$ or $0$, $S$ is the head size, and $\sigma_S : \mathbb{R}^D \to \mathbb{R}^D$ is the Softmax function with output $[\sigma_S(\boldsymbol{x})]_i = e^{\boldsymbol{x}_i} / \sum_{i=1}^{d} e^{\boldsymbol{x}_i}$, $i \in [D]$, for any $\boldsymbol{x} \in \mathbb{R}^D$. Here, $\sigma_S$ operates on each column of the input matrix. For mathematical simplicity, we ignore the layer normalization in attention layers.

The output $\boldsymbol{X} \in \mathbb{R}^{D \times N}$ of the self-attention layer is then passed to the feed-forward layer, given by

$$\mathcal{F}_{FF}(\boldsymbol{X}) := \boldsymbol{X} + \boldsymbol{W}_2 \sigma_R \left[ \boldsymbol{W}_1 \boldsymbol{X} + \boldsymbol{b}_1 \mathbf{1}_N^{\top} \right] + \boldsymbol{b}_2 \mathbf{1}_N^{\top} \in \mathbb{R}^{D \times N},$$

where $\boldsymbol{W}_1 \in \mathbb{R}^{W \times D}$ and $\boldsymbol{W}_2 \in \mathbb{R}^{W \times D}$ are weight matrices with hidden dimension $W$, and $\boldsymbol{b}_1 \in \mathbb{R}^W$, $\boldsymbol{b}_2 \in \mathbb{R}^D$ are bias terms. $\sigma_R$ denotes the element-wise ReLU activation function, i.e., $\sigma_R(x) := \max\{0, x\}$.

The class of Transformer neural networks is then defined as

$$\mathcal{T}(D, H, S, W, L) := \left\{ \mathcal{E}_{out} \circ \mathcal{F}_{FF}^{(L)} \circ \mathcal{F}_{SA}^{(L)} \circ \cdots \circ \mathcal{F}_{FF}^{(1)} \circ \mathcal{F}_{SA}^{(1)} \circ \mathcal{E}_{in} \right\},$$

where $D$ is the embedding size, $H$ is the number of heads, $S$ is the head size, $W$ is the hidden dimension in the feed-forward layers, and $L$ is the number of Transformer layers, each consisting of a self-attention layer and a feed-forward layer. $\mathcal{E}_{in}$ and $\mathcal{E}_{out}$ represent the embedding layer and the decoding layer, respectively, which are two linear affine function. They are designed to change the input and output dimension of Transformers. In this work, we also consider attention-only Transformers, which discard all the feed-forward layers. The class of attention-only Transformer neural networks can be defined as

$$\mathcal{T}(D, H, S, L) := \left\{ \mathcal{E}_{out} \circ \mathcal{F}_{SA}^{(L)} \circ \cdots \circ \mathcal{F}_{SA}^{(1)} \circ \mathcal{E}_{in} \right\}.$$

### 3.2 Feed-Forward Neural Networks

We denote $\mathcal{NN}_\sigma(N, L, \mathbb{R}^d \to \mathbb{R}^{d'})$ as the set of vector-valued functions $\phi : \mathbb{R}^d \to \mathbb{R}^{d'}$ that can be represented by a feed-forward neural network (FFN) with width $\leq N \in \mathbb{N}^+$, depth $\leq L \in \mathbb{N}^+$, and activation function $\sigma$. The width of a FFN refers to the maximum number of neurons in the hidden layers and the depth corresponds to the number of hidden layers. For instance, suppose $\phi : \mathbb{R}^d \to \mathbb{R}^{d'}$ is a vector-valued function realized by a feed-forward neural network with $\sigma$ as the elementwise activation function. Then $\phi$ can be expressed as

$$\phi = \mathcal{L}_L \circ \sigma \circ \mathcal{L}_{L-1} \circ \cdots \circ \sigma \circ \mathcal{L}_1 \circ \sigma \circ \mathcal{L}_0,$$

where each $\mathcal{L}_\ell$ is an affine linear map given by $\mathcal{L}_\ell(\boldsymbol{x}) := \boldsymbol{W}_\ell \boldsymbol{x} + \boldsymbol{b}_\ell$ for $\ell = 0, 1 \cdots, L$. Here, $\boldsymbol{W}_\ell \in \mathbb{R}^{d_{\ell+1} \times d_\ell}$ and $\boldsymbol{b}_\ell \in \mathbb{R}^{d_{\ell+1}}$ are the weight matrix and bias term, respectively, with $d_0 = d$, $d_1, \cdots, d_L \in \mathbb{N}^+$, and $d_{L+1} = d'$. Clearly, $\phi \in \mathcal{NN}_\sigma(N, L, \mathbb{R}^d \to \mathbb{R}^{d'})$ where $N = \max\{d_1, \cdots, d_L\}$.

In terms of the choice of the activation function, we consider $\sigma_R$ (ReLU) and $\sigma_L$ (SiLU) in this work, with definition given in the following

$$\sigma_R(x) := \max\{0, x\}, \quad \sigma_L(x) := \frac{x}{1 + e^{-x}}.$$

It is natural to extend the input of a FFN from vectors to matrices. We redefine each $\mathcal{L}_\ell$ as $\mathcal{L}_\ell(\boldsymbol{X}) := \boldsymbol{W}_\ell \boldsymbol{X} + \boldsymbol{b}_\ell \mathbf{1}_N^{\top}$, where $\boldsymbol{W}_\ell \in \mathbb{R}^{d_{\ell+1} \times d_\ell}$, $\boldsymbol{b}_\ell \in \mathbb{R}^{d_{\ell+1}}$ and $\mathbf{1}_N \in \mathbb{R}^{N \times 1}$ represents the all-1 vector. Clearly, a FFN with matrix input imposes the same operation on each column of the input. In the sequel, we do not distinguish between feed-forward neural networks with vector input and

feed-forward neural networks with matrix input. Similarly, a residual feed-forward neural network $\phi : \mathbb{R}^d \to \mathbb{R}^d$ is defined as follow

$$\phi = \mathcal{L}_L \circ \mathcal{L}_{L-1} \circ \cdots \circ \mathcal{L}_2 \circ \mathcal{L}_1,$$

where each $\mathcal{L}_\ell$ given by $\mathcal{L}_\ell(\boldsymbol{x}) := \boldsymbol{x} + \boldsymbol{W}_\ell^{(2)} \sigma(\boldsymbol{W}_\ell^{(1)} \boldsymbol{x} + \boldsymbol{b}_\ell^{(1)}) + \boldsymbol{b}_\ell^{(2)}$ for $\ell = 1, \cdots, L$. Here, $\boldsymbol{W}_\ell^{(1)} \in \mathbb{R}^{W \times d}$, $\boldsymbol{W}_\ell^{(2)} \in \mathbb{R}^{d \times W}$, and $\boldsymbol{b}_\ell^{(1)} \in \mathbb{R}^W$, $\boldsymbol{b}_\ell^{(2)} \in \mathbb{R}^d$ are the weight matrix and bias term, respectively. Let $\mathcal{NN}_\sigma^{Res}(W, L, \mathbb{R}^d \to \mathbb{R}^d)$ denote the class of residual feed-forward neural networks with $\sigma$ being the activation function.

### 3.3 Formulation of Masked Language modeling

Masked language modeling (MLM) is one of the most widely used pretraining objectives for Transformer-based language models. The key idea is to corrupt an input sequence by replacing some of its tokens with a special placeholder token, denoted by [MASK], and then train the model to recover the original tokens given the corrupted sequence. Moreover, sequences are also augmented with special tokens that serve structural and representational purposes. The token [CLS] is prepended to every sequence and the token [SEP] is appended to mark the end of a sequence, or to separate two sequences in next sentence prediction task. Here, we formally state the assumptions regarding the data.

**Assumption 3.1.** Let $\mathcal{V} \subset \mathbb{R}^D$ be a finite vocabulary and [MASK] denote the mask token with [MASK] $\notin \mathcal{V}$. Each token $\boldsymbol{x} \in \mathcal{V}$ corresponds to a token ID $y \in [C]$, where $C = |\mathcal{V}|$. Let $(\boldsymbol{X}^{(1)}, y^{(1)}), \cdots, (\boldsymbol{X}^{(n)}, y^{(n)}) \subset \mathbb{R}^{D \times N} \times [C]$ be $n$ input-output pairs with $y^{(i)}$ denoting the token ID of the masked token. For any $m_i \in \{1, 2, \cdots, N\}$, the $m_i$-th token in $\boldsymbol{X}^{(i)}$ is masked by the [MASK] token, that is, $\boldsymbol{X}_{:,m_i}^{(i)} = $ [MASK]. We assume that the following conditions are satisfied:

1. There exists $r_1, r_2 > 0$ such that for any $i \in [n]$ and $j \in [N]$, $r_1 \leq \|\boldsymbol{X}_{:,j}^{(i)}\| \leq r_2$.

2. There exists $\delta > 0$ such that for any $i, j \in [n]$ and $k, l \in [N]$, either $\boldsymbol{X}_{:,k}^{(i)} = \boldsymbol{X}_{:,l}^{(j)}$ or $\|\boldsymbol{X}_{:,k}^{(i)} - \boldsymbol{X}_{:,l}^{(j)}\| \geq \delta$ holds.

3. For any $i, j \in [n]$, $\boldsymbol{X}^{(i)} \neq \boldsymbol{X}^{(j)}$ up to permutations.

4. There are no duplicated tokens in each $\boldsymbol{X}^{(i)}$ for $i \in [n]$.

We provide a discussion on Assumption 3.1 in Appendix C, which shows that Assumption 3.1 is mild and holds in a wide range of practical settings. With Assumption 3.1 in hand, we define the masked language modeling task as follows.

**Definition 3.1** (Mased Language Modeling). Let $(\boldsymbol{X}^{(1)}, y^{(1)}), \cdots, (\boldsymbol{X}^{(n)}, y^{(n)}) \subset \mathbb{R}^{D \times N} \times [C]$ be $n$ input-output pairs, satisfying Assumption 3.1. The goal of masked language modeling is to find a Transformer $\mathcal{F}$ such that $\mathcal{F}(\boldsymbol{X}^{(i)})_{:,m_i} \approx y^{(i)}$ for any $i \in [n]$, which can be formulated as: Given any $\varepsilon > 0$, there exists a Transformer $\mathcal{F} \in \mathcal{T}(D, H, S, W, L)$ for some $D, H, S, W, L \in \mathbb{Z}^+$ such that

$$\left| \mathcal{F}(\boldsymbol{X}^{(i)})_{:m_i} - y^{(i)} \right| < \varepsilon \quad \text{for any } i \in [n].$$

Note that Definition 3.1 describes a simplified variant of the standard masked language modeling task. In practice, approximately 15% of tokens in the sequence are selected for prediction, 80% of which are replaced by [MASK], 10% by a random token, and 10% remain unchanged. In constrast, out setting considers only the case where exactly one token is masked. Furthermore, in real-world models, the output of the Transformer is passed through a linear layer, and the model is trained by minimizing the cross-entropy loss. When the number of data points $n$ and the vocabulary size $C$ are large, this linear layer becomes limited to predict accurately the original tokens. Instead, we consider the setup by letting the output correspond directly to the token ID of the masked token. Finally, our formulation takes a constructive perspective, in the sense that we explicitly design a Transformer that satisfies the desired properties.

# 4    POSITION-WISE NON-LINEARITY

In vanilla Transformers, the primary source of non-linearity comes from the feed-forward layers, which apply column-wise non-linear transformations to each token independently. By contrast, self-attention layers are not explicitly designed to provide such position-wise non-linearity. Each token is updated as a weighted linear combination of linearly transformed token representations, and the only non-linear components in the self-attention mechanism are the dot-product similarity and the subsequent Softmax normalization, which serve to compute attention weights rather than to transform token representations directly.

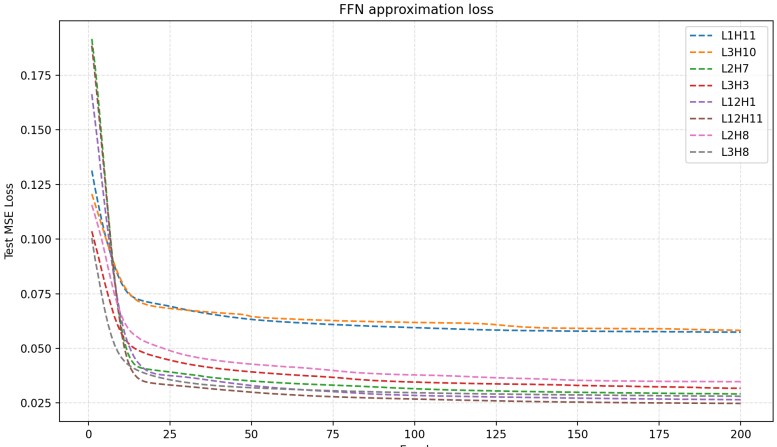

Figure 2: We train a feed-forward layer with hidden dimension 16 to approximate the attention heads mentioned in Figure 1 using a knowledge distillation method. The result shows that the approximation error of the Head 1 in Layer 1 and Head 10 in Layer 3 is higher than others, suggesting that their behavior deviates from that of a feed-forward layer. In contrast, the remaining heads primarily capture the interaction between data tokens and special tokens either [CLS] or [SEP], exhibiting behavior more similar to a feed-forward layer.

In this section, we show that by introducing a fixed, global token into the input sequence (similar to [CLS] or [SEP] tokens in Bert (Devlin et al., 2019)) and a well-designed mask matrix, which prevents the unnecessary interaction among tokens, the Softmax operation in self-attention can act analogously to an activation function and self-attention layers can approximate deep feed-forward neural networks. Under this construction, self-attention layers gain the ability to implement position-wise non-linearity, thereby broadening their functional role beyond contextual interactions. The following Theorem 4.1 summarizes our results.

**Theorem 4.1** (Approximation of feed-forward neural networks). *For any $\varepsilon > 0$, $M > 0$ and any FFN $\boldsymbol{f} \in \mathcal{NN}_{\sigma_R}(W, L, \mathbb{R}^D \to \mathbb{R}^{D'})$, there exists an attention-only Transformer $\mathcal{F} \in \mathcal{T}(\max\{W, D\} + 2, \max\{W, D\} + 1, 1, L)$ such that*

$$\|\mathcal{F}(\boldsymbol{X}) - \boldsymbol{f}(\boldsymbol{X})\|_\infty < \varepsilon \quad \textit{for any } \boldsymbol{X} \in [-M, M]^{D \times N}.$$

*The embedding layer in $\mathcal{F}$ is defined by*

$$\mathcal{E}_{in}(\boldsymbol{X}) := \begin{pmatrix} \boldsymbol{X} & \boldsymbol{0} \\ \mathbf{1}_{1 \times N} & \boldsymbol{0} \\ \boldsymbol{0}_{(W-D)^+ \times N} & \boldsymbol{0} \\ \boldsymbol{0} & 1 \end{pmatrix} \in \mathbb{R}^{(\max\{W, D\}+2) \times (N+1)}.$$

The proof of Theorem 4.1 is postponed to Appendix E. Note that after the embedding layer, an extra vector $\begin{pmatrix} \boldsymbol{0} & \cdots & \boldsymbol{0} & 1 \end{pmatrix}^\top$ is appended to the sequence, which acts as a global bias. With a special design of the mask matrix, we ensure that each data token only interacts with itself and this bias vector and we let the value vector (i.e. $\boldsymbol{W}_V \boldsymbol{x}$) of this bias become $\boldsymbol{0}$, meaning that this bias vector contributes no information to each data token, but only bias the attention weights such that it can be regarded as a activation function. Let's take the case $N = 2$ as an example: suppose that

$\boldsymbol{X} = (\boldsymbol{x}_1 \quad \boldsymbol{x}_2) \in \mathbb{R}^{D \times 2}$, the output of a self-attention layer without skip connection given input $\boldsymbol{X}$ is $(a_1 \boldsymbol{W}_V \boldsymbol{x}_1 + a_2 \boldsymbol{W}_V \boldsymbol{x}_2 \quad \boldsymbol{0})$, where $a_1, a_2$ are attention weights and we only let $\boldsymbol{x}_1$ interact with $\boldsymbol{x}_1$ and $\boldsymbol{x}_2$ by using a mask matrix. If $\boldsymbol{W}_V \boldsymbol{x}_2 = \boldsymbol{0}$, we have $a_1 \boldsymbol{W}_V \boldsymbol{x}_1 + a_2 \boldsymbol{W}_V \boldsymbol{x}_2 = a_1 \boldsymbol{W}_V \boldsymbol{x}_1$, meaning that $\boldsymbol{x}_2$ contributes nothing to $\boldsymbol{x}_1$ but only bias its attention weight. Note that $a_1 \boldsymbol{W}_V \boldsymbol{x}_1$ is a non-linear Transformation of $\boldsymbol{x}_1$ due to the Softmax function contained in $a_1$.

**Remark 4.1.** In Figure 1, we observe that some attention heads in **Bert** (12 layers with 12 heads in each layer) have a special attention pattern, in which the data tokens mostly interacts with [CLS] or [SEP]. Moreover, in Figure 10, we observe that the norm of the value vector of [CLS] or [SEP] is small in certain layers. As a result, in Figure 2, we train a FFN with hidden dimension 16 to mimic these attention heads, finding that their behavior is closer to that of a FFN. Since some works (Melas-Kyriazi, 2021; Bozic et al., 2023; Zhang et al., 2020) proposed to replace some self-attention heads by feed-forward layers, our result may serve as a possible selection standard to determine which heads can be substituted.

**Remark 4.2.** We obseve that Theorem 4.1 is similar to prompt tuning (Lester et al., 2021) in several aspects. In the task of prompt tuning, the parameters of the model are kept frozen and the parameters in prompts are trainable. By contrast, in Theorem 4.1, the parameters of the model are trainable (depend on the feed-forward neural network to be approximated), while the appended vector remains fixed. Another correspondence lies in the training stage: prompt tuning is applied after pre-training the model while Theorem 4.1 is realized during pre-training. The results of (Nakada et al., 2025) can therefore be regarded as the prompt tuning counterpart of ours. One additional insight is that one could relax our construction by making the appended vector into trainable or by extending it to a longer sequence of vectors, similar to the case in prompt tuning.

The following corollary demonstrates that the appended vector can be selected arbitrarily, subject only to a mild constraint.

**Corollary 4.1.** *For any $\varepsilon > 0$, $M > 0$ and any FFN $\boldsymbol{f} \in \mathcal{NN}_{\sigma_R}(W, L, \mathbb{R}^D \to \mathbb{R}^{D'})$, there exists an attention-only Transformer $\mathcal{F} \in \mathcal{T}(\max\{W, D\} + 2, \max\{W, D\} + 1, 1, L)$ such that*

$$\|\mathcal{F}(\boldsymbol{X}) - \boldsymbol{f}(\boldsymbol{X})\|_\infty < \varepsilon \quad \text{for any } \boldsymbol{X} \in [-M, M]^{D \times N}.$$

*The embedding layer in $\mathcal{F}$ is defined by*

$$\boldsymbol{\mathcal{E}}_{in}(\boldsymbol{X}) := \begin{pmatrix} \boldsymbol{X} \\ \boldsymbol{1}_{1 \times N} & \boldsymbol{v} \\ \boldsymbol{0}_{(W-D)^+ \times N} \\ \boldsymbol{0} \end{pmatrix} \in \mathbb{R}^{(\max\{W, D\} + 2) \times (N+1)},$$

*where $\boldsymbol{v} \in \mathbb{R}^{\max\{W, D\} + 2}$ is an arbitrary vector with the last element $\boldsymbol{v}_{-1} \neq 0$.*

## 5 CONTEXTUAL NON-LINEARITY

Due to the combined effects of Softmax function and dot product, the self-attention mechanism induces a non-linear mapping from token representations to attention weights. Importantly, the non-linearity is not applied to tokens in isolation but rather arises from interaction among tokens within the same sequence. Since the attention weights are instance-dependent, each input will result in a different attention pattern. We refer to this ability to generate dynamic, input-specific attention patterns as **contextual non-linearity**. In contrast, we define the contextual linearity as the case where the attention pattern is fixed and does not vary with the input. In this setting, the way tokens attend to each other is purely linear, as the attention weights remain constant across all input sequences. This motivates the following definition of a self-attention layer with a constant attention pattern.

**Definition 5.1** (Self-attention with a Fixed Attention Pattern)**.** Given any matrix $\boldsymbol{A} \in \mathbb{R}^{N \times N}$ with each element $\boldsymbol{A}_{i,j} > 0$. An self-attention layer $\mathcal{F}_{SA}^{\boldsymbol{A}}$ with attention pattern $\boldsymbol{A}$ and $H$ heads is defined as

$$\mathcal{F}_{SA}^{\boldsymbol{A}}(\boldsymbol{X}) := \boldsymbol{X} + \sum_{i=1}^{H} \boldsymbol{W}_O \boldsymbol{W}_V \boldsymbol{X} \boldsymbol{A}.$$

In the remainder of this work, we allow Transformers to include both standard self-attention layers and fixed-pattern self-attention layers. We will use the superscript to distinguish between the two.

In masked language modeling, there are several tokens masked by a special token denoted by [MASK], which is universal across all input sequences. Transformers are trained to predict the original token before being masked based on the contexts in which it appears. This is a task requiring Transformers to distinguish the same token in different contexts because the tokens before being masked can be different. This is a concept first proposed by (Yun et al., 2019; Kim et al., 2022; Kajitsuka & Sato, 2023). In Figure 3, we trace the cosine similarity between the [MASK] tokens in different contexts across layers. It is shown that the representation of [MASK] tokens becomes more and more distinguishable as the layer goes deeper. To see the ability to differentiate tokens of

Figure 3: **Cosine similarity between** [MASK] **tokens in different contexts.** We trace the cosine similarity between [MASK] tokens in **Bert** (12 layers with 12 heads in each layer). In blue line, we consider two similar contexts, while in red line, we consider two totally different contexts. All [MASK] tokens are placed in the first position of the sequence and the sequence length is chosen to be the same to eliminate the influence brought by position. The result shows that the difference between the [MASK] representation becomes large as the depth goes deeper, especially when the contexts are significantly different.

contextual non-linearity, the following proposition proves that one standard self-attention layer with one head can distinguish the [MASK] in different contexts.

**Proposition 5.1** (Standard Self-attention). *For any input-label pairs* $(\boldsymbol{X}^{(1)}, y^{(1)}), \cdots, (\boldsymbol{X}^{(n)}, y^{(n)})$ *satisfying Assumption 3.1, there exists a self-attention layer* $\boldsymbol{\mathcal{F}}_{SA} \in \mathcal{T}(D, 1, 1, 1)$ *such that*

$$\boldsymbol{\mathcal{F}}_{SA}(\boldsymbol{X}^{(i)})_{:,m_i} \neq \boldsymbol{\mathcal{F}}_{SA}(\boldsymbol{X}^{(j)})_{:,m_j} \quad \text{for any } i \neq j \in [n].$$

The proof technique basically follows (Kajitsuka & Sato, 2023), in which we resort to the separation ability of Boltzmann operator and its close relationship with Softmax function. Then, it is natural to ask that whether a self-attention layer with a fixed attention pattern remains this property. The following proposition provides a deterministic answer.

**Proposition 5.2** (Fixed Attention Pattern). *For any input-label pairs* $(\boldsymbol{X}^{(1)}, y^{(1)}), \cdots, (\boldsymbol{X}^{(n)}, y^{(n)})$ *satisfying Assumption 3.1, there exists a self-attention layer* $\boldsymbol{\mathcal{F}}_{SA}^{\boldsymbol{A}} \in \mathcal{T}(D, 1, 1, 1)$ *with a fixed attention pattern* $\boldsymbol{A}$ *such that*

$$\boldsymbol{\mathcal{F}}_{SA}^{\boldsymbol{A}}(\boldsymbol{X}^{(i)})_{:,m_i} \neq \boldsymbol{\mathcal{F}}_{SA}^{\boldsymbol{A}}(\boldsymbol{X}^{(j)})_{:,m_j} \quad \text{for any } i \neq j \in [n].$$

Note that the attention pattern $\boldsymbol{A}$ in Proposition 5.2 is trainable, meaning that it depends on the training dataset $(\boldsymbol{X}^{(1)}, y^{(1)}), \cdots, (\boldsymbol{X}^{(n)}, y^{(n)})$. So, the total number of parameters becomes $O(N^2)$ instead of $O(DN)$ in Proposition 5.1. In the following, we consider any randomly chosen attention pattern, which is independent of the training dataset. Our result shows that any attention pattern is as powerful as trainable one with the help of a feed-forward layer.

**Proposition 5.3** (Random Attention Pattern). *For any input-label pairs* $(\boldsymbol{X}^{(1)}, y^{(1)}), \cdots, (\boldsymbol{X}^{(n)}, y^{(n)})$ *satisfying Assumption 3.1, and any attention pattern* $\boldsymbol{A} \in \mathbb{R}^{N \times N}$ *with* $\boldsymbol{A}_{i,j} > 0$ *for any* $i, j \in [N]$, *there exists a Transformer* $\boldsymbol{\mathcal{F}} = \boldsymbol{\mathcal{E}}_{out} \circ \boldsymbol{\mathcal{F}}_{SA}^{\boldsymbol{A}} \circ \boldsymbol{\mathcal{F}}_{FF} \circ \boldsymbol{\mathcal{E}}_{in} \in \mathcal{T}(\max\{3(n-1)N, D\}, 1, 1, 3(n-1)N, 1)$ *such that*

$$\boldsymbol{\mathcal{F}}(\boldsymbol{X}^{(i)})_{:,m_i} \neq \boldsymbol{\mathcal{F}}(\boldsymbol{X}^{(j)})_{:,m_j} \quad \text{for any } i \neq j \in [n].$$

All the proofs of this section are postponed to Appendix F. In the proof of Proposition 5.3, the feed-forward layer maps each token into a standard basis in a higher space, whose dimension depends on the total number of different tokens in the vocabulary. That is why the hidden dimension scales with the number of tokens $(n-1)N$. According to the results in (Vardi et al., 2021; Kajitsuka & Sato, 2024), if we use a deep feed-forward neural network with constant width, the depth only grows as $O(\sqrt{nN})$, which is much more parameter-efficient. As shown in Figure 3, the difference between the representation of the same [MASK] token in distinct contexts becomes larger as the layer goes deeper. However, in Proposition 5.3, we only use one self-attention layer and one feed-forward layer to distinguish [MASK] tokens.

## 6 MASKED LANGUAGE MODELING

As an application, we consider masked language modeling with attention-only Transformers. In Definition 3.1, we need to predict the masked tokens based on its contexts. Since the [MASK] token is universal across corrupted sequences, we first use one self-attention layer to map each [MASK] token to distinct vectors, and then use another self-attention layer to implement point fitting. Our results are summarized in the following theorem.

**Theorem 6.1.** *For any $\varepsilon > 0$ and $n$ input-label pairs $(\boldsymbol{X}^{(1)}, y^{(1)}), \cdots, (\boldsymbol{X}^{(n)}, y^{(n)}) \subset \mathbb{R}^{D \times N} \times [C]$, which satisfy the Assumption 3.1, there exists an attention-only Transformer $\boldsymbol{\mathcal{F}} \in \mathcal{T}(\max\{3n, D\} + 2, \max\{3n, D\} + 1, 1, 2)$ such that*

$$|\boldsymbol{\mathcal{F}}(\boldsymbol{X}^{(i)})_{:,m_i} - \boldsymbol{y}^{(i)}| < \varepsilon \quad \text{for any } i = 1, \cdots, n.$$

The proof of Theorem 6.1 is in Appendix G. Theorem 6.1 shows that a two-layer attention-only Transformer can accurately predict the masked tokens to any precision. The first Transformer layer is constructed from Proposition 5.1, which distinguishes the same [MASK] token in different contexts by mapping them to different values. The second self-attention layer is augmented with a fixed vector attended to the input and approximately implement a FFN from Lemma H.4, which implements "key-value memory". The input represents the key and it is mapped to the corresponding value. To be specific, for $n$ key-value pairs, $\{\boldsymbol{x}^{(1)} \to y^{(i)}, \cdots, \boldsymbol{x}^{(n)} \to y^{(i)}\}$ with $\boldsymbol{x}^{(i)} \in \mathbb{R}^D$, $y^{(i)} \in \mathbb{R}$ and $\boldsymbol{x}^{(i)} \neq \boldsymbol{x}^{(j)}$ for any $i \neq j \in [n]$, there exsits a FFN $\boldsymbol{f} \in \mathcal{NN}_{\sigma_R}(3n, 1, \mathbb{R}^D \to \mathbb{R})$ such that

$$\boldsymbol{f}(\boldsymbol{x}^{(i)}) = y^{(i)} \quad \text{for any } i \in [n].$$

Note that it is natural to restate Theorem 6.1 based on Proposition 5.2 and 5.3, where we only need to modify the first self-attention layer. As a result, all attention patterns in this Transformer are independent of input sequences.

## 7 CONCLUSION

In this paper, we study the position-wise and contextual non-linearity of attention mechanism. Firstly, we prove that by augmenting the input with a fixed bias vector, a stack of self-attention layers are able to approximate deep feed-forward neural networks. This demonstrates that attention layers alone are capable of implementing position-wise non-linearity. We also identify some attention heads in real-world pre-trained language models, which perform similar behavior to that of a FFN. Furthermore, we prove that the instance-dependent interaction patterns of self-attention are not essential for distinguishing the same token in different contexts, calling into question the necessity of contextual non-linearity in certain settings. By integrating both position-wise and contextual non-linearity, we prove that a two-layer attention-only Transformer suffices to make accurate predictions in masked language modeling. Finally, our empirical findings and validation support our theory.

**Limitation:** Sparsity and low-rank condition have not been considered in the construction of attention patterns in Proposition 5.2 and 5.3. Moreover, studying the gradient dynamic during masked language modeling would offer a complementary perspective for our constructive results and yield further insights. Since our work only focuses the pre-training process, fine-tuning or prompt tuning and the performance on downstream tasks should also be studied.

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

# Appendix

## A Notation Table

**Functions**

| | |
|---|---|
| $\boldsymbol{f} : \mathbb{A} \to \mathbb{B}$ | The function $f$ with domain $\mathbb{A}$ and range $\mathbb{B}$ |
| $\boldsymbol{f} \circ \boldsymbol{g}$ | Composition of the functions $f$ and $g$ |
| $\boldsymbol{f}(\boldsymbol{x}; \boldsymbol{\theta})$ | A function of $\boldsymbol{x}$ parametrized by $\boldsymbol{\theta}$. (Sometimes we write $f(\boldsymbol{x})$ and omit the argument $\boldsymbol{\theta}$ to lighten notation) |
| $\sigma_R(x)$ | ReLU function, $\max\{x, 0\}$ |
| $\sigma_L(x)$ | SiLU function, $\dfrac{x}{1 + e^{-x}}$ |
| $\sigma_S(\boldsymbol{x})$ | Softmax function, $\sigma_S(\boldsymbol{x})_i = \dfrac{\exp(\boldsymbol{x}_i)}{\sum_{i=1}^{d} \exp(\boldsymbol{x}_i)}$ |
| $\|\boldsymbol{x}\|_p$ | $L^p$ norm of $\boldsymbol{x}$ |
| $\|\boldsymbol{x}\|$ | $L^2$ norm of $\boldsymbol{x}$ |
| $\|\boldsymbol{x}\|_\infty$ | $\infty$ norm of $\boldsymbol{x}$ |
| $x^+$ | Positive part of $x$, i.e., $\max(0, x)$ |
| $\boldsymbol{\mathcal{F}}_{SA}$ | Standard self-attention layer |
| $\boldsymbol{\mathcal{F}}_{FF}$ | Feed-forward layer |
| $\boldsymbol{\mathcal{F}}_{SA}^{\boldsymbol{A}}$ | Self-attention layer with fixed attention pattern $\boldsymbol{A}$ |
| $\mathcal{T}(D, H, S, W, L)$ | Transformer neural network class with embedding size $D$, number of heads $H$, head size $S$, hidden dimension $W$, number of layers $L$ |
| $\mathcal{T}(D, H, S, L)$ | Attention-only Transformer neural network class with embedding size $D$, number of heads $H$, head size $S$, number of layers $L$ |
| $\mathcal{NN}_\sigma(W, L, \mathbb{R}^d \to \mathbb{R}^{d'})$ | Feed-forward neural network class with width $W$, depth $L$, input dimension $d$ and output dimension $d'$, activation function $\sigma$ |
| $\mathcal{NN}_\sigma^{Res}(W, L, \mathbb{R}^d \to \mathbb{R}^d)$ | Residual feed-forward neural network class with hidden dimension $W$, number of layers $L$, input dimension and output dimension $d$, activation function $\sigma$ |

**Numbers and Arrays**

| | |
|---|---|
| $a$ | A scalar (integer or real) |
| $\boldsymbol{a}$ | A vector |
| $\boldsymbol{A}$ | A matrix |
| $\boldsymbol{I}_n$ | Identity matrix with $n$ rows and $n$ columns |
| $\boldsymbol{I}$ | Identity matrix with dimensionality implied by context |
| $\boldsymbol{e}_i$ | Standard basis vector $[0, \ldots, 0, 1, 0, \ldots, 0]$ with a 1 at position $i$ |
| $\boldsymbol{1}_{n \times m}$ | All-one matrix with dimensionality $n \times m$ |

**Sets**

| | |
|---|---|
| $\mathbb{R}$ | The set of real numbers |
| $\mathbb{R}^D$ | The set of $D$-dimensional real vectors |
| $\mathbb{R}^D_{>0}$ | The set of $D$-dimensional positive real vectors |
| $\{0, 1\}$ | The set containing 0 and 1 |
| $\{0, 1, \ldots, n\}$ | The set of all integers between 0 and $n$ |
| $[n]$ | The set of all integers between 1 and $n$, that is, $[n] = \{1, \cdots, n\}$ |

**Indexing**

| | |
|---|---|
| $\boldsymbol{a}_i$ | Element $i$ of vector $\boldsymbol{a}$, with indexing starting at 1 |
| $\boldsymbol{a}_{-1}$ | The last element of vector $\boldsymbol{a}$ |
| $\boldsymbol{A}_{i,j}$ | Element $i, j$ of matrix $\boldsymbol{A}$ |
| $\boldsymbol{A}_{i,:}$ | Row $i$ of matrix $\boldsymbol{A}$ |
| $\boldsymbol{A}_{:,i}$ | Column $i$ of matrix $\boldsymbol{A}$ |

**Asymptotics**

| | |
|---|---|
| $f(n) = O(g(n))$ | $f$ grows at most as fast as $g$ for sufficiently large $n$ |
| $f(n) = \tilde{O}(g(n))$ | $f$ grows at most as fast as $g$ for sufficiently large $n$, up to logarithmic factors |
| $f(n) = \Omega(g(n))$ | $f$ grows at least as fast as $g$ for sufficiently large $n$ |
| $f \lesssim g$ | There exists a positive constant $c$ such that $f \leq cg$ holds |

## B  ADDITIONAL RELATED WORK

**Theoretical Understanding of Transformers.**  One the fundamental study on the expressive capacity of Transformer architecture is to explore its approximation ability, that is, investgating whether Transformers can approximate functions that belong to a given function class. The most seminal work by (Yun et al., 2019) provided the first universal approximation theorem for Transformers, showing that any continuous sequence-to-sequence functions defined on a compact domain can be approximated by Transformer to any precision. They also extended the results to sparse Transformers in (Yun et al., 2020). Gurevych et al. (2022) gave a constructive method to show that Transformers can approximate piecewise polynomials. Jiang & Li (2024) built their results of approximating continuous functions by shallow Transformers based on the Kolmogorov Representation Theorem. Takakura & Suzuki (2023) provided both approximation and estimation error with $\gamma$-smooth function class under the assumption that the input is infinite dimensional. Similarly, Havrilla & Liao (2024) leveraged manifold hypothesis, assuming that the input data has a low-dimensional structure and established approximation results for $\beta$-Hölder continuous functions. Kajitsuka & Sato (2023) showed that Transformers with one single-head self-attention layer can be a universal approximator, where they dug deeply into the relationship between the softmax function in the self-attention layer and the Boltzmann operator. Takeshita & Imaizumi (2025) proved that Transformers can efficiently approximate column-symmetric polynomials with respect to the number of parameters. Concurrently, Jiao et al. (2025a) established the approximation results of Transformers for Hölder class and Sobolev class under $L^p$-norm, where $p \in [0, +\infty]$. Besides, their another work (Jiao et al., 2025b) proved that Transformers are able to overcome the curse of dimensionality based on Kolmogorov-Arnold Representation Theorem. Hu et al. (2025) tried to avoid the dependence on large ReLU feed-forward layers by proving that attention layers alone can approximate a generalized version of ReLU function and hence subsumes any known approximators based on ReLU feed-forward neural networks. Similarly, Liu et al. (2025) proved that a single self-attention layer, preceded by sum-

of-linear transformations, is capable of approximating any continuous functions on a compact domain under $L^\infty$-norm, highlighting the inherent expressive power of attention mechanism alone. Cheng et al. (2025) investigated the universal approximation property of Transformer-type architectures, providing a unified theoretical framework that incorporates various architecture variants.

As for Transformers with prompts, Wang et al. (2023) showed that prompt tuning Transformers can be a universal approximator to Lipschitz sequence-to-sequence functions and Hu et al. (2024) further extended their results to Transformers with only one self-attention layer. Petrov et al. (2024) proved that prompt tuning Transformers is able to approximate sequence-to-sequence functions defined on the hypersphere. Besides, in Nakada et al. (2025), they proved that a fixed-size Transformer with well-designed prompts can exactly compute a certain class of ReLU feed-forward neural networks.

Another line of theoretical study on the expressive ability of Transformers is to see whether Transformers can achieve zero loss on finite input-output pairs. Kim et al. (2022) is the first work to study the memorization capacity of Transformers. They proved that Transformers with $\tilde{O}(d + n + \sqrt{nN})$ parameters are able to memorize length $n$ and $d$-dimensional input tokens. Mahdavi et al. (2023) showed that a multi-head self-attention mechanism with $H$ heads and $O(Hd^2)$ parameters is capable of memorizing $O(Hn)$ data samples. Kajitsuka & Sato (2023) proved that a Transformer with only one single-head self-attention layer have data memorization ability. Chen & Zou (2024) built the results of Transformers with ReLU activation function under the assumption that each data label is distinct. Kajitsuka & Sato (2024) established their results with an optimal number of parameters. It was shown that Transformers with $\tilde{O}(\sqrt{N})$ parameters in the sequence-to-sequence prediction setting and $\tilde{O}(\sqrt{nN})$ parameters in the sequence-to-sequence prediction task.

## C  A DISCUSSION ON ASSUMPTION 3.1

In this section we provide a discussion on Assumption 3.1, which shows that it is mild and can be easily satisfied in real-world settings.

**Assumption C.1** (Restatement of Assumption 3.1). Let $\mathcal{V} \subset \mathbb{R}^D$ be a finite vocabulary and $[\text{MASK}]$ denote the mask token with $[\text{MASK}] \notin \mathcal{V}$. Each token $\boldsymbol{x} \in \mathcal{V}$ corresponds to a token ID $y \in [C]$, where $C = |\mathcal{V}|$. Let $(\boldsymbol{X}^{(1)}, y^{(1)}), \cdots, (\boldsymbol{X}^{(n)}, y^{(n)}) \subset \mathbb{R}^{D \times N} \times [C]$ be $n$ input-output pairs with $y^{(i)}$ denoting the token ID of the masked token. For any $m_i \in \{1, 2, \cdots, N\}$, the $m_i$-th token in $\boldsymbol{X}^{(i)}$ is masked by the $[\text{MASK}]$ token, that is, $\boldsymbol{X}^{(i)}_{:,m_i} = [\text{MASK}]$. We assume that the following conditions are satisfied:

1. There exists $r_1, r_2 > 0$ such that for any $i \in [n]$ and $j \in [N]$, $r_1 \le \|\boldsymbol{X}^{(i)}_{:,j}\| \le r_2$.

2. There exists $\delta > 0$ such that for any $i, j \in [n]$ and $k, l \in [N]$, either $\boldsymbol{X}^{(i)}_{:,k} = \boldsymbol{X}^{(j)}_{:,l}$ or $\|\boldsymbol{X}^{(i)}_{:,k} - \boldsymbol{X}^{(j)}_{:,l}\| \ge \delta$ holds.

3. For any $i, j \in [n]$, $\boldsymbol{X}^{(i)} \neq \boldsymbol{X}^{(j)}$ up to permutations.

4. There are no duplicated tokens in each $\boldsymbol{X}^{(i)}$ for $i \in [n]$.

In this following, we provide a step-by-step interpretation of Assumption 3.1, showing that our assumption on data is reasonable and easily satisfied in real-world scenarios:

1. This $r$ naturally exists in real-world datasets, since every data point is stored with finite precision. Besides, due to the widely utilized normalization technique during training (i.e., layer normalization, batch normalization) or during preprocessing, every data point is well-bounded.

2. Since we consider a discrete dataset, which contains finite data points, so this $\delta$ inherently exists. This assumption does not impose any extra limitation on the data, but only provides convenience for theoretical analysis.

3. Since there exists one token in each $\boldsymbol{X}^{(i)}$ being masked by $[\text{MASK}]$, it is possible that there exits two data points $\boldsymbol{X}^{(i)}$ and $\boldsymbol{X}^{(j)}$, which are not equal to each other before being masked but equal to each other up to permutation after it. Moreover, It is widely known that

Transformer architecture is permutation equivariant without special positional encoding, that is, given any permutation matrix $\boldsymbol{P}$, $f(\boldsymbol{XP}) = f(\boldsymbol{X})\boldsymbol{P}$ holds for any Transformer $f$. As a result, $f(\boldsymbol{X}^{(i)})$ is also a permutation of $f(\boldsymbol{X}^{(j)})$, meaning that $f(\boldsymbol{X}^{(i)})_{:,m_i} \equiv f(\boldsymbol{X}^{(j)})_{:,m_j}$, which contradicts with fact that $y^{(i)}$ is possible to be different from $y^{(j)}$. In natural languages, permutation equivalence of words is not common. Different permutation usually leads to different meanings, which motivates us to assume that no permutated data points are in our setting.

4. In practical scenarios, it is possible that two exactly same tokens appear in one sequence. However, the permutation equivariant limitation of Transformers makes the two tokens undistinguishable. Many works (Su et al., 2024; Ke et al., 2020) have been working on designing effective positional encoding to break this limit. In this work, since we do not focus on positional encoding, it is reasonable that we consider there are no duplicated tokens in each $\boldsymbol{X}^{(i)}$.

# D    EXPERIMENTS

This section provides experimental results to back up our theory. We validate that (i) Self-attention augmented with a fixed vector can approximate a FFN effectively (Theorem 4.1), (ii) a two layer attention-only Transformer and its variants can perform well in masked language modeling (Theorem 6.1, Proposition 5.2, Proposition 5.3). We conduct all experiments using one NVIDIA A100 GPU. Our code is based on standard PyTorch modules.

## D.1    PROOF-OF-CONCEPT EXPERIMENTS ON THEOREM 4.1

**Objective: Verifying self-attention augmented with a fixed vector approximates a FFN.** We investigate accuracy of self-attention with a fixed vector appended in the input approximating a $FFN$, comparing with standard self-attention.

**Data Generation and Model Architecture.** We randomly generate $\boldsymbol{X} \in \mathbb{R}^{D \times N}$ drawn from a normal distribution, where $\boldsymbol{X} \sim N(0,1)$. The number of training data is 75 and test data is 25. We initialize the self-attention layer with $D = 16$, $N = 8$, $S = 1$, $H = 75$, $L = 3$ where $D$ is the embedding size, $N$ is the sequence length, $S$ is the head size, $H$ is the number of heads and $L$ is the number of self-attention layers. Furthermore, we also randomly generate a FFN $\boldsymbol{f} \in \mathcal{NN}_{\sigma_R}(W, L, \mathbb{R}^D \to \mathbb{R}^D)$ as the target, with $W = H = 75$, $L = 3$. The batch size is chosen to be 1 and optimizer is Adam. Following the original Theorem 4.1, we exclude dropout and layer normalization in our experiments following our theory.

**Results.** We let $\mathbf{S}$ denote the standard self-attention and $\mathbf{F}$ represents the self-attention with a fixed vector. As shown in Figure 4, evaluated on $log$ test MSE error, we can see that $\mathbf{F}$ achieves smaller error than $S$, which proves our theory.

**Sensitivity of Approximation to the Number of Heads.** We study the relationship between the approximation error and the number of heads. As shown in Figure 5, increasing the number of heads, the approximation error of $\mathbf{F}$ decreases stably and reach equilibrium, while the error curve of $\mathbf{S}$ fluctuates as the number of heads growing.

**Sensitivity of Approximation to the Number of Layers.** We consider to study the approximation ability between the number of self-attention layers. In Theorem 4.1, the required number of self-attention layers equal to the depth of the FFN to be approximated. It is natural to ask that can fewer layers achieve the similar results or can more layers achieve better performance? Fixing a FFN with depth 5, we use a stacked self-attention with layer $\{1, 2, \cdots, 10\}$ to approximate it. As shown in Figure 6, the minimum of $\mathbf{F}$ is achieved when the number of layers is 5, which proves our theory.

## D.2    PROOF-OF-CONCEPT EXPERIMENTS ON THEOREM 6.1

**Objective: Verifying the Effectiveness in Masked Language Modeling.** We investigate whether four different models that are based on Theorem 6.1, Proposition 5.2 and 5.3, can handle the masked language modeling task.

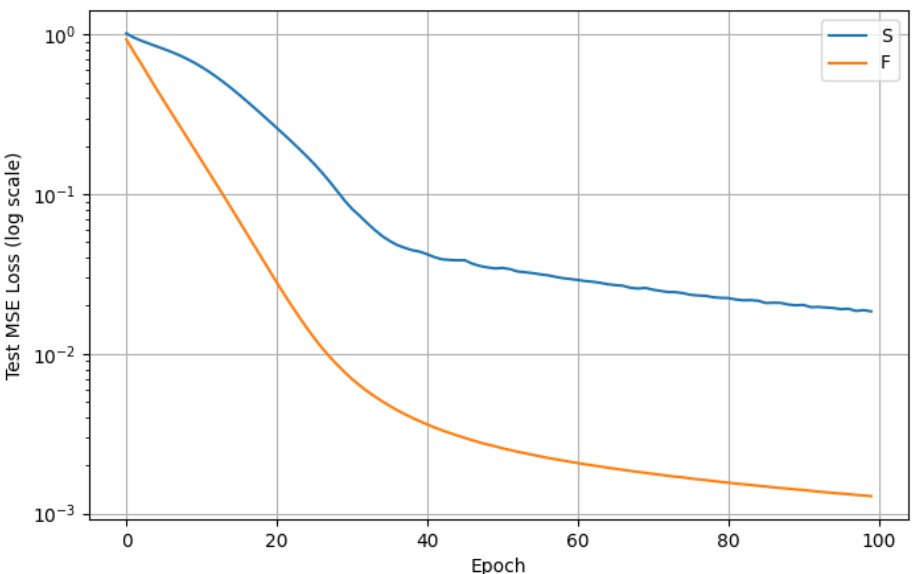

Figure 4: **Test Loss.** We report the test error of both **F** and **S** versus the number of training epochs. We use synthetic data of 75 training data points and 25 testing data points, with sequence length being 8 and embedding dimension 16. We set the batch size to be 16. The optimizer used is Adam with learning rate 0.001. The result shows that after training for 100 epochs, **F** can effectively approximate the target FFN while **S** can not.

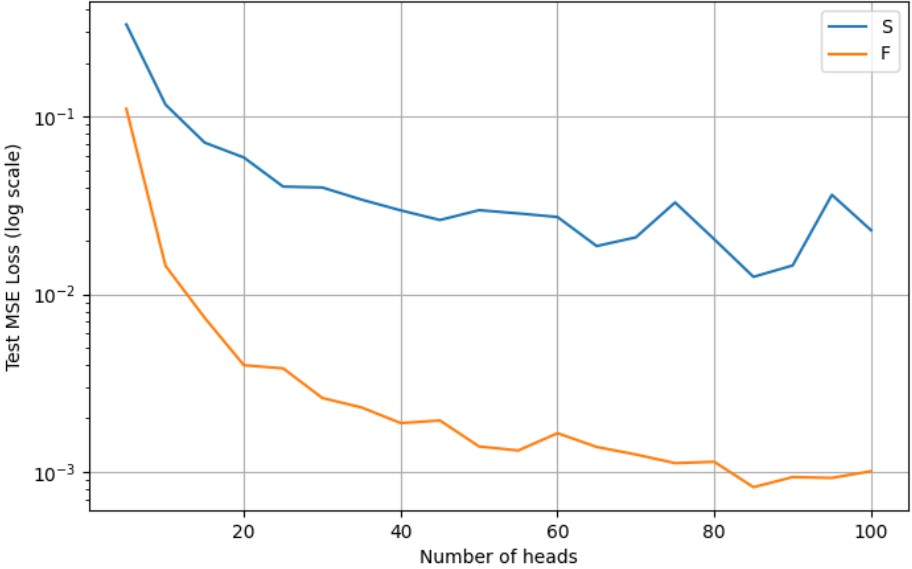

Figure 5: **Test Loss of different number of heads.** We report the test error of both **F** and **S** with number of heads being chosen from $\{5, 10, \cdots, 100\}$. We use synthetic data of 75 training data points and 25 testing data points, with sequence length being 8 and embedding dimension 16. We set the batch size to be 16. The optimizer used is Adam with learning rate 0.001. The result shows that after training for 100 epochs, increasing the number of heads, **F** can effectively approximates the target FFN, while the error of **S** remains far above zero.

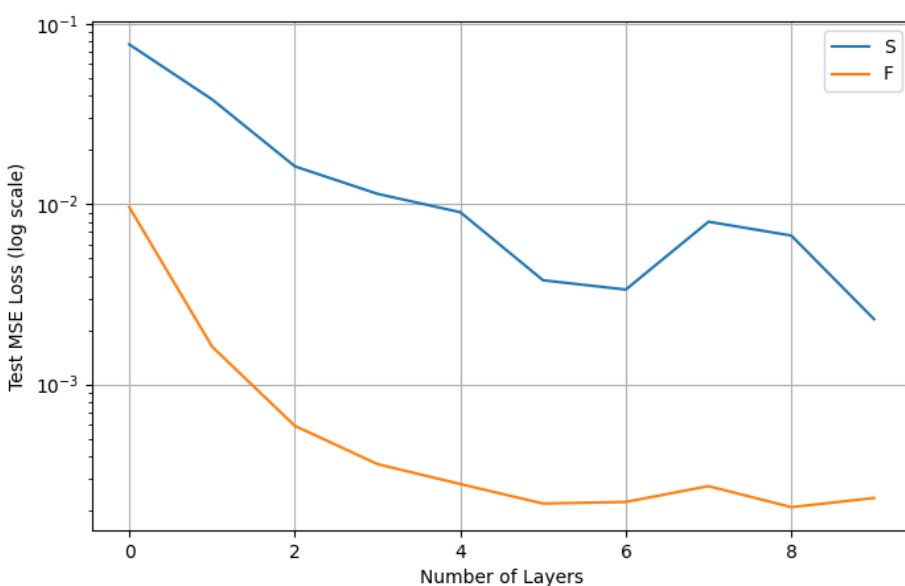

Figure 6: **Test Loss of different number of layers.** We report the test error of both **F** and **S** approximating a FFN with depth 5 versus the number of self-attention layers $\{1, 2, \cdots, 10\}$. We use synthetic data of 75 training data points and 25 testing data points, with sequence length being 8 and embedding dimension 16. We set the batch size to be 16. The optimizer used is Adam with learning rate 0.001. The result shows that after training for 100 epochs, the minimum error of **F** is achieved at the 5 layer.

**Data Generation and Model Architecture.** We let the vocabulary size $C$ be 2048, number of training data points is 1024, sequence length 8, embedding size 16. we ensure that there are no repeated tokens in a single sequence even without positional encoding. We consider four models:

1. **S**: A standard Transformer, consisting of one self-attention layer and one feed-forward layer (Baseline).

2. **F**: **S** + replace the feed-forward layer by a self-attention layer augmented by a fixed vector (Theorem 6.1).

3. **F+R**: **F** + replace the self-attention layer with a randomly chosen, non-trainable attention pattern + add an extra layer of self-attention augmented by a fixed vector (Proposition 5.3).

4. **F+T**: **F** + replace the self-attention layer with a trainable and input-independent attention pattern (Proposition 5.2).

Note that We let the number of heads equals to the number of training data points just as the setting in Theorem 6.1, the head size is chosen to be 1. Similarly, we design a mask that only allows the interaction between data tokens and the fixed vector in the second self-attention layer. The batch is 128 and the optimizer is Adam. We only focus on the training loss curve.

**Results.** As shown in Figure 7, we can see that these four models are able to predict accurately in masked language modeling task, by achieving nearly zero loss. Interestingly, **F+T** and **F+R** own a faster convergence rate than that of **S** and **F**, meaning that a randomly chosen or a trainable, universal attention pattern is already powerful.

**Ablation on the extra self-attention layer in F+R.** Note that in Proposition 5.3, we show that a random attention pattern with a feed-forward layer can do well in token distinction task. In the model **F+R**, we use a self-attention with a fixed vector to replace the feed-forward layer. Now, we consider getting rid of this extra self-attention layer, and see how the performance will change. We denote the

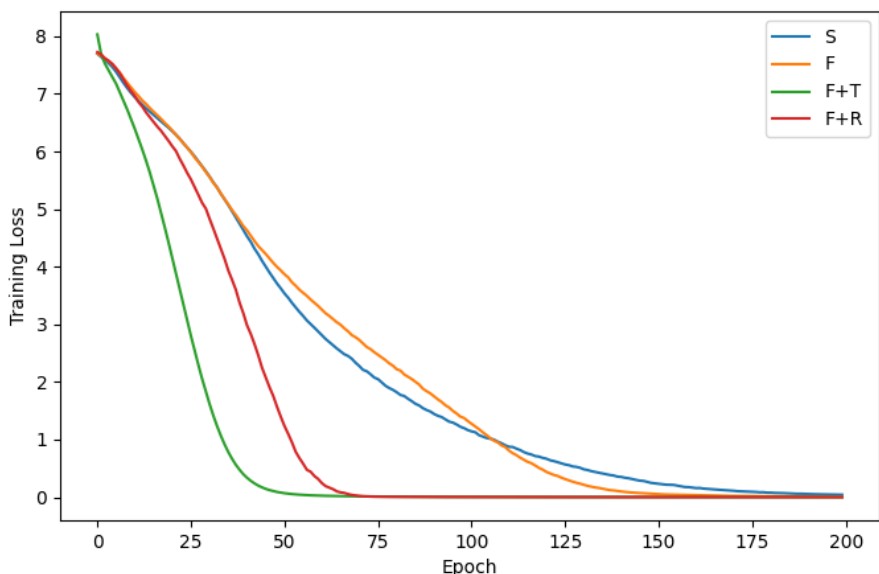

Figure 7: **Training Loss.** We report the training error of four models: **S**, **F**, **F+T**, **F+R** in masked language modeling. We use synthetic data of 1024 training data points, with sequence length being 8 and embedding dimension 16. We set the batch size to be 128. The optimizer used is Adam with learning rate 0.001. The result shows that after training for 200 epochs, all four models achieve nearly zero loss while **F+T** and **F+R** converge much faster.

model by **F+R+noF**. As shown in Figure 8, **F+R+noF** can also reach nearly zero loss only with a slower convergence rate. We leave its theoretical foundation to future reasearch.

**The Linear dependence of Number of heads on the number of training data points.** In Theorem 6.1, we prove that the required number of heads grows linearly as the number of training data points increases. We set the number of the training data points to be $\{512, 1024, \cdots, 4608\}$, and the number of heads to be $\{32, 64, \cdots, 608\}$, and see what is the minimum number of heads needed to achieve a training loss less than 0.05 on these training data points. As shown in Figure 9, there is a clear linear relationship between the number of training data points and the number of heads needed to achieve a small loss, which validates our theory.

**Small norm of value vectors.** Note that in the proof of Theorem 4.1, we need the value vector (i.e., $W_V x$) of the bias vector appended to the input equals $\mathbf{0}$. We observe that this phenomenon happens in real-world pre-trained language models. Although some attention heads intend to assign large attention weights to special tokens, such as [CLS] or [SEP], the value vector of them are relatively small. We consider tracing the norm of the vector value through layers and results are shown in Figure 10.

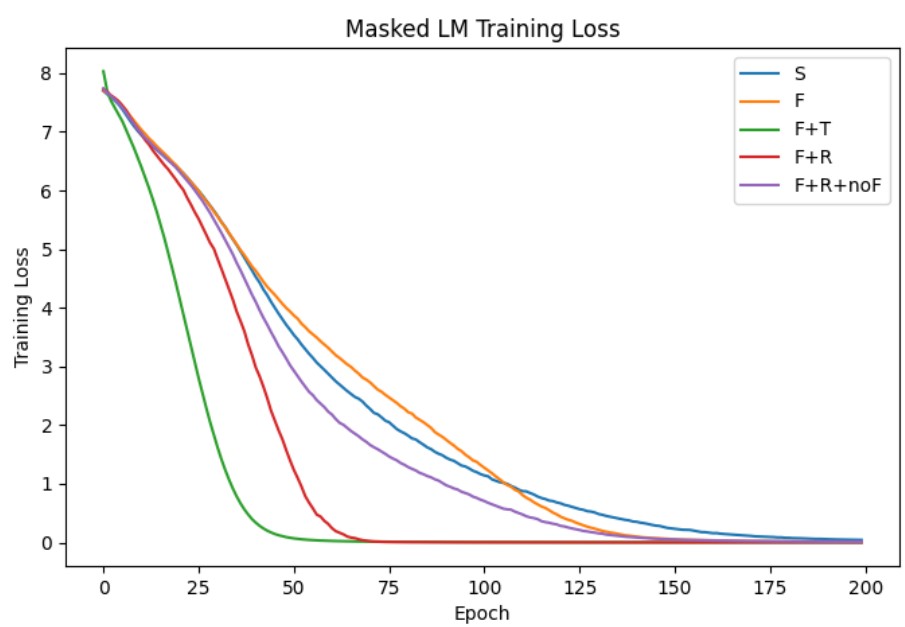

Figure 8: **Training Loss.** We report the training error of five models: **S**, **F**, **F+T**, **F+R** and **F+R+noF** in masked language modeling. We use synthetic data of 1024 training data points, with sequence length being 8 and embedding dimension 16. We set the batch size to be 128. The optimizer used is Adam with learning rate 0.001. The result shows that after training for 200 epochs, **F+R+noF** has a slower convergence rate than that of **F+R** but still can achieve nearly zero training loss.

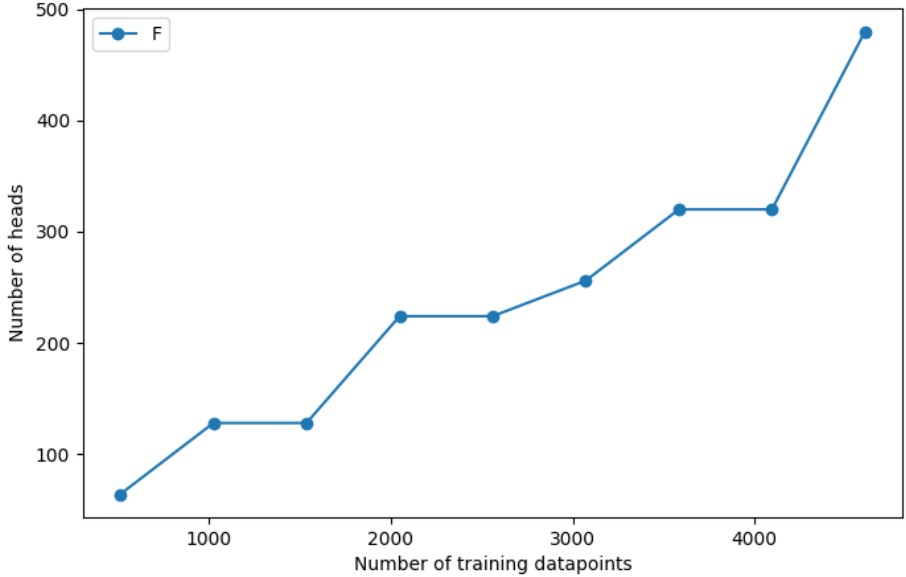

Figure 9: **Number of heads needed to achieve zero loss under different number of training data points.** We report the number of heads required for producing accurate prediction in masked language modeling. We consider model **F**. We use synthetic data of $\{512, 1024, \cdots, 4608\}$ training data points, with sequence length being 8 and embedding dimension 16. We set the batch size to be 128. The optimizer used is Adam with learning rate 0.001. The result shows that after training for 200 epochs, there is an approximate linear trend in the number of heads.

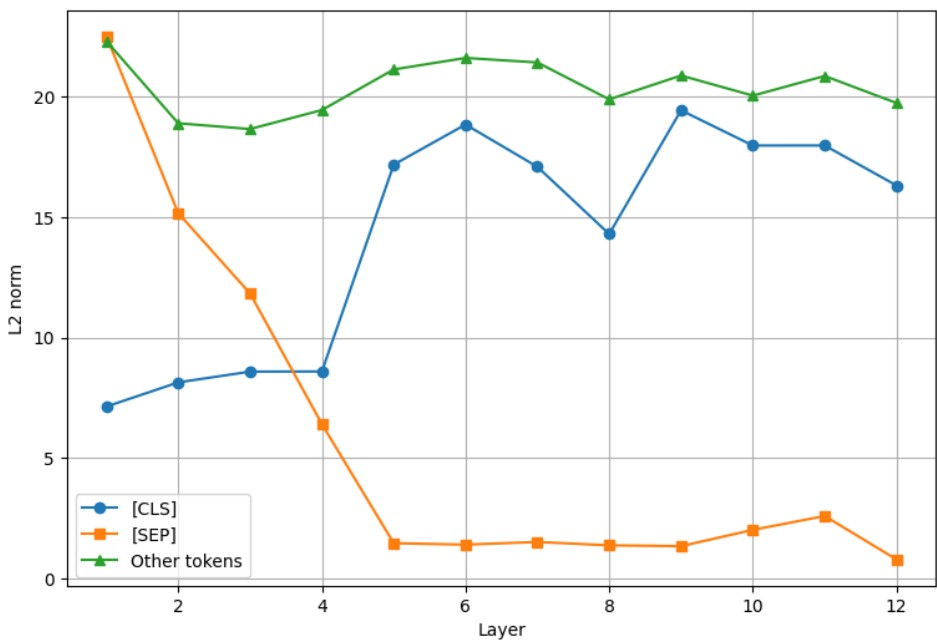

Figure 10: **Norm of the value vectors of different tokens.** We report the norm of the value vector of different tokens across layers. We consider **Bert** (12 layers and 12 heads in each layer). The norm is computed averagely over 100 randomly chosen data points in SST-2 dataset. The result shows the norm of value vector of [CLS] is small before the fourth layer and the norm of value vector of [SEP] becomes nearly zero after layer 5.

## E    PROOF OF SECTION 4

### E.1    PROOF OF THEOREM 4.1

**Theorem E.1** (Restatement of Theorem 4.1). *For any $\varepsilon > 0$, $M > 0$ and any FFN $\boldsymbol{f} \in \mathcal{NN}_{\sigma_R}(W, L, \mathbb{R}^D \to \mathbb{R}^{D'})$, there exists an attention-only Transformer $\boldsymbol{\mathcal{F}} \in \mathcal{T}(\max\{W, D\} + 2, \max\{W, D\} + 1, 1, L)$ such that*

$$\|\boldsymbol{\mathcal{F}}(\boldsymbol{X}) - \boldsymbol{f}(\boldsymbol{X})\|_\infty < \varepsilon \quad \text{for any } \boldsymbol{X} \in [-M, M]^{D \times N}.$$

*The embedding layer in $\boldsymbol{\mathcal{F}}$ is defined by*

$$\boldsymbol{\mathcal{E}}_{in}(\boldsymbol{X}) := \begin{pmatrix} \boldsymbol{X} & \boldsymbol{0} \\ \boldsymbol{1}_{1 \times N} & \boldsymbol{0} \\ \boldsymbol{0}_{(W-D)^+ \times N} & \boldsymbol{0} \\ \boldsymbol{0} & 1 \end{pmatrix} \in \mathbb{R}^{(\max\{W, D\}+2) \times (N+1)}.$$

*Proof of Theorem 4.1.* **Step 1:** In this step, We show that each $\sigma_R$ or $\sigma_L$ activated FFN with bias term in each layer can be transformed into a no-bias FFN by slightly increasing the width, which is summarized in the following Lemma.

**Lemma E.1.** *Let $\sigma = \sigma_R$ or $\sigma_L$. For any $\boldsymbol{f} \in \mathcal{NN}_\sigma(W, L, \mathbb{R}^d \to \mathbb{R}^{d'})$, there exists $\boldsymbol{g} \in \mathcal{NN}_\sigma(W + 1, L, \mathbb{R}^d \to \mathbb{R}^{d'})$ with no bias terms in each layer such that*

$$\boldsymbol{f}(\boldsymbol{X}) = \boldsymbol{g}(\widehat{\boldsymbol{X}}),$$

*where $\widehat{\boldsymbol{X}} = \begin{pmatrix} \boldsymbol{X} \\ \boldsymbol{1}_{1 \times N} \end{pmatrix}$.*

The proof of Lemma E.1 is postponed to Appendix E.2.

**Step 2:** We build the connection between FFNs activated by $\sigma_L$ and FFNs activated by $\sigma_R$. The following Lemma demonstrates that for any $\sigma_R$-activated FFN, there exists a $\sigma_L$-activated FFN can approximates it to any precision.

**Lemma E.2.** *for any $\varepsilon > 0$, $M > 0$, and FFN without bias $\boldsymbol{f} \in \mathcal{NN}_{\sigma_R}(W, L, \mathbb{R}^D \to \mathbb{R}^{D'})$, there exists a FFN without bias $\boldsymbol{g} \in \mathcal{NN}_{\sigma_L}(W, L, \mathbb{R}^D \to \mathbb{R}^{D'})$ such that*

$$\|\boldsymbol{f}(\boldsymbol{X}) - \boldsymbol{g}(\boldsymbol{X})\|_{\sup([-M,M]^{D \times N})} < \varepsilon.$$

The proof of Lemma E.2 is placed in Appendix E.3.

**Step 3:** In this step, we aim to build the connection between residual FFNs and non-residual FFNs. Specifically, the Lemma below shows that each non-residual FFN can be represented by a residual FFN when the activation function is $\sigma_L$.

**Lemma E.3.** *For any FFN without bias terms $\boldsymbol{f} \in \mathcal{NN}_{\sigma_L}(W, L, \mathbb{R}^D \to \mathbb{R}^{D'})$ with $D' \leq W$, there exists a residual FFN $\boldsymbol{g} \in \mathcal{NN}_{\sigma_L}^{Res}(\max\{W, D\}, L, \mathbb{R}^{\max\{W,D\}} \to \mathbb{R}^{\max\{W,D\}})$ such that for any $\boldsymbol{X} \in \mathbb{R}^{D \times N}$, the following holds*

$$g(\widehat{\boldsymbol{X}}) = \begin{pmatrix} \boldsymbol{f}(\boldsymbol{X}) \\ \boldsymbol{0} \end{pmatrix} \in \mathbb{R}^{\max\{W,D\} \times N},$$

*where* $\widehat{\boldsymbol{X}} = \begin{pmatrix} \boldsymbol{X} \\ \boldsymbol{0}_{(W-D)^+ \times N} \end{pmatrix} \in \mathbb{R}^{\max\{W,D\} \times N}$.

The proof of Lemma E.3 is postponed to Appendix E.4.

**Step 4:** In this part, we build a close connection between self-attention and FFNs activated by $\sigma_L$. The following Lemma shows that we can use a single self-attention layer to implement one layer FFN activated by $\sigma_L$.

**Lemma E.4.** *for any $\boldsymbol{W}_1 \in \mathbb{R}^{W \times D}$, $\boldsymbol{W}_2 \in \mathbb{R}^{D \times W}$ and $\boldsymbol{X} \in \mathbb{R}^{D \times N}$, there exists a self-attention layer $\mathcal{F}_{SA} \in \mathcal{T}(D+1, W, 1, 1)$ such that*

$$\mathcal{F}_{SA}(\widehat{\boldsymbol{X}}) = \begin{pmatrix} \boldsymbol{W}_2 \sigma_L(\boldsymbol{W}_1 \boldsymbol{X}) + \boldsymbol{X} & \boldsymbol{0} \\ \boldsymbol{0} & 1 \end{pmatrix} \in \mathbb{R}^{(D+1) \times (N+1)},$$

*where* $\widehat{\boldsymbol{X}} = \begin{pmatrix} \boldsymbol{X} & \boldsymbol{0} \\ \boldsymbol{0} & 1 \end{pmatrix} \in \mathbb{R}^{(D+1) \times (N+1)}$.

The proof of Lemma E.4 is places in Appendix E.5.

**Step 5:** In this part, we show that for any residual feed-forward neural network without bias terms $\boldsymbol{f} \in \mathcal{NN}_{\sigma_L}^{Res}(W, L, \mathbb{R}^D \to \mathbb{R}^D)$, there exists an attention-only Transformer $\mathcal{F} \in \mathcal{T}(D+1, W, 1, L)$ such that

$$\mathcal{F}(\widehat{\boldsymbol{X}}) = \begin{pmatrix} \boldsymbol{f}(\boldsymbol{X}) & \boldsymbol{0} \\ \boldsymbol{0} & 1 \end{pmatrix} \in \mathbb{R}^{(D+1) \times (N+1)},$$

where $\widehat{\boldsymbol{X}} = \begin{pmatrix} \boldsymbol{X} & \boldsymbol{0} \\ \boldsymbol{0} & 1 \end{pmatrix} \in \mathbb{R}^{(D+1) \times (N+1)}$.

According to the definition of residual neural networks, $\boldsymbol{f}$ can be written as

$$\boldsymbol{f} = \mathcal{L}_L \circ \cdots \circ \mathcal{L}_1,$$

where $\mathcal{L}_\ell(\boldsymbol{X}) = \boldsymbol{X} + \boldsymbol{W}_\ell^{(2)} \sigma_L\left(\boldsymbol{W}_\ell^{(1)} \boldsymbol{X}\right)$ with $\boldsymbol{W}_\ell^{(2)} \in \mathbb{R}^{D \times W}$, $\boldsymbol{W}_\ell^{(1)} \in \mathbb{R}^{W \times D}$.

Through the analysis in **step 1**, there exists a self-attention layer $\mathcal{F}_{SA}^{(\ell)} \in \mathcal{T}(D+1, W, 1, 1)$ such that

$$\mathcal{F}_{SA}^{(\ell)}(\widehat{\boldsymbol{X}}) = \begin{pmatrix} \boldsymbol{W}_\ell^{(2)} \sigma_L(\boldsymbol{W}_\ell^{(1)} \boldsymbol{X}) + \boldsymbol{X} & \boldsymbol{0} \\ \boldsymbol{0} & 1 \end{pmatrix} \in \mathbb{R}^{(D+1) \times (N+1)},$$

Let $\mathcal{F} := \mathcal{F}_{SA}^{(L)} \circ \cdots \circ \mathcal{F}_{SA}^{(1)}$. Since the 1 in the bottom right corner is kept in the whole residual flow, we can verify that

$$\begin{aligned} \mathcal{F}(\widehat{\boldsymbol{X}}) &= \mathcal{F}_{SA}^{(L)} \circ \cdots \circ \mathcal{F}_{SA}^{(1)}(\widehat{\boldsymbol{X}}) \\ &= \mathcal{F}_{SA}^{(L)} \circ \cdots \circ \mathcal{F}_{SA}^{(2)} \begin{pmatrix} \boldsymbol{W}_1^{(2)} \sigma_L(\boldsymbol{W}_1^{(1)} \boldsymbol{X}) + \boldsymbol{X} & \boldsymbol{0} \\ \boldsymbol{0} & 1 \end{pmatrix} \\ &\quad \vdots \\ &= \begin{pmatrix} \boldsymbol{f}(\boldsymbol{X}) & \boldsymbol{0} \\ \boldsymbol{0} & 1 \end{pmatrix} \in \mathbb{R}^{(D+1) \times (N+1)}. \end{aligned}$$

The proof of **step 5** is completed by noting that $\mathcal{F} \in \mathcal{T}(D+1, W, 1, L)$.

**Step 6:** Putting everything together.

For any $\boldsymbol{f} \in \mathcal{NN}_{\sigma_R}(W, L, \mathbb{R}^D \to \mathbb{R}^{D'})$, there exists a no-bias $\boldsymbol{f}_1 \in \mathcal{NN}_{\sigma_R}(W+1, L, \mathbb{R}^{D+1} \to \mathbb{R}^{D'})$ such that

$$\boldsymbol{f}_1(\boldsymbol{X}_1) = \boldsymbol{f}(\boldsymbol{X}),$$

where $\boldsymbol{X}_1 = \begin{pmatrix} \boldsymbol{X} \\ \boldsymbol{1}_{1 \times N} \end{pmatrix} \in \mathbb{R}^{(D+1) \times N}$. Moreover, according to Lemma E.2, for any $\varepsilon > 0$, there exists $\boldsymbol{f}_2 \in \mathcal{NN}_{\sigma_L}(W+1, L, \mathbb{R}^{D+1} \to \mathbb{R}^{D'})$ such that

$$\| \boldsymbol{f}_2(\boldsymbol{X}_1) - \boldsymbol{f}_1(\boldsymbol{X}_1) \|_{\sup([-M,M]^{D' \times N})} < \varepsilon.$$

By applying Lemma E.3, there exists a residual FFN

$$\boldsymbol{f}_3 \in \mathcal{NN}_{\sigma_L}(\max\{W+1, D+1\}, L, \mathbb{R}^{\max\{W+1, D+1\}} \to \mathbb{R}^{\max\{W+1, D+1\}})$$

such that

$$\boldsymbol{f}_3(\boldsymbol{X}_2) = \begin{pmatrix} \boldsymbol{f}_2(\boldsymbol{X}_1) \\ \boldsymbol{0}_{(W-D)^+ \times N} \end{pmatrix},$$

where $\boldsymbol{X}_2 = \begin{pmatrix} \boldsymbol{X}_1 \\ \boldsymbol{0}_{(W-D)^+ \times N} \end{pmatrix} \in \mathbb{R}^{(\max\{W, D\}+1) \times N}$. Finally, Lemma 4.1 and **Step 5** show that there exists an attention-only Transformer $\mathcal{F}_{SA} \in \mathcal{T}(\max\{W, D\}+2, \max\{W, D\}+1, 1, L)$ such that

$$\mathcal{F}_{SA}(\boldsymbol{X}_3) = \begin{pmatrix} \boldsymbol{f}_3(\boldsymbol{X}_2) & \boldsymbol{0} \\ \boldsymbol{0} & 1 \end{pmatrix} \in \mathbb{R}^{(\max\{W, D\}+2) \times (N+1)},$$

where $\boldsymbol{X}_3 = \begin{pmatrix} \boldsymbol{X}_2 & \boldsymbol{0} \\ \boldsymbol{0} & 1 \end{pmatrix} \in \mathbb{R}^{(\max\{W, D\}+2) \times (N+1)}$. The proof is completed by defining $\mathcal{E}_{in}$ and $\mathcal{E}_{out}$ as

$$\mathcal{E}_{in}(\boldsymbol{X}) = \begin{pmatrix} \boldsymbol{X} & \boldsymbol{0} \\ \boldsymbol{1}_{1 \times N} & \boldsymbol{0} \\ \boldsymbol{0}_{(W-D)^+ \times N} & \boldsymbol{0} \\ \boldsymbol{0} & 1 \end{pmatrix} \in \mathbb{R}^{(\max\{W, D\}+2) \times (N+1)} \quad \text{for any } \boldsymbol{X} \in \mathbb{R}^{D \times N},$$

$$\mathcal{E}_{out}(\boldsymbol{X}) = \boldsymbol{X}_{1:D, 1:N}.$$

$\square$

### E.2 PROOF OF LEMMA E.1

*Proof of Lemma E.1.* We first consider the case when $\sigma = \sigma_R$. According to the definition of FFNs, $\boldsymbol{f}$ can be written as

$$\boldsymbol{f} = \boldsymbol{\mathcal{L}}_L \circ \sigma_R \cdots \circ \sigma_R \circ \boldsymbol{\mathcal{L}}_0,$$

where each $\boldsymbol{\mathcal{L}}_\ell$ is given by $\boldsymbol{\mathcal{L}}_\ell(\boldsymbol{x}) := \boldsymbol{W}_\ell \boldsymbol{x} + \boldsymbol{b}_\ell$ for $\ell = 0, 1 \cdots, L$ with $\boldsymbol{W}_\ell \in \mathbb{R}^{D_{\ell+1} \times D_\ell}$, $\boldsymbol{b}_\ell \in \mathbb{R}^{D_{\ell+1}}$, and $D_0 = D, D_1, \cdots, D_L \in \mathbb{N}^+$, and $D_{L+1} = D'$.

Let $\widehat{\boldsymbol{X}} = \begin{pmatrix} \boldsymbol{X} \\ \boldsymbol{1}_{1 \times N} \end{pmatrix}$, where $\boldsymbol{1}_{1 \times N}$ is the all-1 vector of size $1 \times N$. We define

$$\widehat{\boldsymbol{W}}_\ell = \begin{pmatrix} \boldsymbol{W}_\ell & \boldsymbol{b}_\ell \\ \boldsymbol{0} & 1 \end{pmatrix} \in \mathbb{R}^{(D_{l+1}+1) \times (D_l+1)}, \quad \text{for any } \ell = 0, \cdots, L-1,$$

$$\widehat{\boldsymbol{W}}_L = \begin{pmatrix} \boldsymbol{W}_L & \boldsymbol{b}_L \end{pmatrix} \in \mathbb{R}^{D' \times (D_L+1)}.$$

Let $\boldsymbol{g} = \widehat{\boldsymbol{\mathcal{L}}}_L \circ \sigma_R \circ \cdots \circ \sigma_R \circ \widehat{\boldsymbol{\mathcal{L}}}_0$. Through direct verification, we have

$$\boldsymbol{g}(\widehat{\boldsymbol{X}}) = \boldsymbol{f}(\boldsymbol{X}).$$

Then ,we consider when $\sigma = \sigma_L$. Let $s$ be the solution of the equation $1 = \frac{x}{1+e^{-x}}$ and let $\widehat{\boldsymbol{X}} = \begin{pmatrix} \boldsymbol{X} \\ \boldsymbol{1}_{1 \times N} \cdot \end{pmatrix}$. Similarly, we define

$$\widehat{\boldsymbol{W}}_\ell = \begin{pmatrix} \boldsymbol{W}_\ell & \boldsymbol{b}_\ell \\ \boldsymbol{0} & s \end{pmatrix} \in \mathbb{R}^{(D_{l+1}+1) \times (D_l+1)}, \quad \text{for any } \ell = 0, \cdots, L-1,$$

$$\widehat{\boldsymbol{W}}_L = \begin{pmatrix} \boldsymbol{W}_L & \boldsymbol{b}_L \end{pmatrix} \in \mathbb{R}^{D' \times (D_L+1)}.$$

Let $\boldsymbol{g} = \widehat{\boldsymbol{\mathcal{L}}}_L \circ \sigma_R \circ \cdots \circ \sigma_R \circ \widehat{\boldsymbol{\mathcal{L}}}_0$ Through direct verification, we have

$$\boldsymbol{g}(\widehat{\boldsymbol{X}}) = \boldsymbol{f}(\boldsymbol{X}),$$

which completes the proof. □

### E.3 PROOF OF LEMMA E.2

This is proof is an extension of Zhang et al. (2024), in which they consider the bias term in each layer.

*Proof of Lemma E.2.* It is straightforward to verify that

$$\frac{\sigma_L(\eta \cdot x)}{x} \to \sigma_R(x) \quad \text{as } \eta \to 0^+ \quad \text{for any } x \in [-M, M].$$

Note that $\frac{\sigma_L(\eta \cdot x)}{x}$ can be implemented by a 1-layer and 1-width FFN activated by $\sigma_L$. Assume that $\boldsymbol{f}$ can be represented in the following form

$$\boldsymbol{f} = \boldsymbol{\mathcal{L}}_L \circ \sigma_R \cdots \circ \sigma_R \circ \boldsymbol{\mathcal{L}}_0,$$

where each $\boldsymbol{\mathcal{L}}_\ell$ is given by $\boldsymbol{\mathcal{L}}_\ell(\boldsymbol{x}) := \boldsymbol{W}_\ell \boldsymbol{x}$ for $\ell = 0, 1 \cdots, L$ with $\boldsymbol{W}_\ell \in \mathbb{R}^{D_{\ell+1} \times D_\ell}$, and $D_0 = D$, $D_1, \cdots, D_L \in \mathbb{N}^+$, and $D_{L+1} = D'$, $\max\{D_1, \cdots, D_L\} \leq W$. Let $\sigma_{L,\eta} = \frac{\sigma_L(\eta \cdot x)}{x}$, we define

$$\phi_\eta(\boldsymbol{x}) := \boldsymbol{\mathcal{L}}_L \circ \sigma_{L,\eta} \circ \cdots \circ \sigma_{L,\eta} \circ \boldsymbol{\mathcal{L}}_0 \quad \text{for any } \boldsymbol{x} \in \mathbb{R}^D.$$

It is easy to verify that

$$\phi_\eta \in \mathcal{NN}_{\sigma_L}(W, L, \mathbb{R}^D \to \mathbb{R}^{D'}),$$

and $\phi_\eta$ does not have bias terms. Later, we prove there exists $\eta = \eta_0$ such that

$$\|\phi_\eta - f\|_{\sup([-M,M]^D)} \to 0 \quad \text{as } \eta \to 0^+ \quad \text{for any } x \in [-M, M]^D.$$

for $\ell = 1, \cdots, L+1$, we define

$$h_\ell(x) := \mathcal{L}_{\ell-1} \circ \sigma_R \circ \mathcal{L}_{\ell-2} \circ \cdots \circ \sigma_R \circ \mathcal{L}_1 \circ \sigma_R \circ \mathcal{L}_0(x),$$

and

$$h_{\ell,\eta}(x) := \mathcal{L}_{\ell-1} \circ \sigma_{L,\eta} \circ \mathcal{L}_{\ell-2} \circ \cdots \circ \sigma_{L,\eta} \circ \mathcal{L}_1 \circ \sigma_{L,\eta} \circ \mathcal{L}_0(x).$$

It is clear that $h_\ell$ and $h_{\ell,\eta}$ are mappings from $\mathbb{R}^D$ to $\mathbb{R}^{D_\ell}$ for $\ell = 1, \cdots, L+1$.

For $\ell = 1, \cdots, L+1$, we prove by induction that

$$\|h_{\ell,\eta} - h_\ell\|_{\sup([-M,M]^D)} \to 0 \quad \text{as } \eta \to 0^+ \quad \text{for any } x \in [-M, M]^D. \tag{E.1}$$

First, we consider the case $\ell = 1$. Clearly,

$$h_{1,\eta} = \mathcal{L}_0 = h_1(x),$$

which means that Equation (E.4) holds for $\ell = 1$.

Next, supposing Equation $E.4$ holds for $\ell = i \in \{1, \cdots, L\}$, we aim to prove that is also holds for $\ell = i + 1$. Determine $R > 0$ via

$$R = \sup \left\{ \|h_j(x)\|_{\ell^\infty} + 1 : x \in [-M, M]^D, \quad j = 1, 2, \cdots, L+1 \right\},$$

where the continuity of $\sigma_R$ guarantees the above supremum is finite. By the induction hypothesis, we have

$$\|h_{i,\eta} - h_i\|_{\sup([-M,M]^D)} \to 0 \quad \text{as } \eta \to 0^+ \quad \text{for any } x \in [-M, M]^D. \tag{E.2}$$

Since for any $x \in [-M, M]^D$, we have $\|h_i(x)\|_\infty \leq M$ and

$$\|h_{i,\eta}(x)\|_\infty \leq \|h_i(x)\|_\infty + 1 \leq M \quad \text{for small } \eta > 0.$$

Recall that $\sigma_{L,\eta}(t) \to \sigma_R(t)$ as $\eta \to 0^+$ for any $t \in [-R, R]$. Then, we have

$$\|\sigma_{L,\eta} \circ h_{\ell,\eta}(x) - \sigma_R \circ h_{\ell,h}(x)\|_{\sup([-M,M]^D)} \to 0 \quad \text{as } \eta \to 0^+ \quad \text{for any } x \in [-M, M]^D. \tag{E.3}$$

Due to the continuity of $\sigma_R$, we deduce

$$\|\sigma_R \circ h_{i,\eta}(x) - \sigma_R \circ h_i(x)\|_{\sup([-M,M]^D)} \to 0 \quad \text{as } \eta \to 0^+ \quad \text{for any } x \in [-M, M]^D. \tag{E.4}$$

Therefore, for any $x \in [-M, M]^D$, as $\eta \to 0^+$, we have

$$\begin{aligned} \sigma_{L,\eta} &\circ h_{i,\eta}(x) - \sigma_R \circ h_i(x) \\ &= \sigma_{L,h} \circ h_{i,\eta}(x) - \sigma_R \circ h_{i,\eta}(x) + \sigma_R \circ h_{i,\eta}(x) - \sigma_R \circ h_i(x) \to 0 \end{aligned}$$

implying

$$\|h_{i+1,\eta}(x) - h_{i+1}(x)\|_{\sup([-M,M]^D)} = \|\mathcal{L}_i \circ \sigma_{L,\eta} \circ h_{i,\eta} - \mathcal{L}_i \circ \sigma_R \circ h_i\|_{\sup([-M,M]^D)} \to 0,$$

which means that Equation E.4 holds for $\ell = i + 1$. So we completes the inductive step.

By the principle of induction, as $\eta \to 0^+$ and for any $x \in [-M, M]^D$ we have

$$\|\phi_\eta - f\|_{\sup([-M,M]^D)} = \|h_{L+1,\eta} - h_{L+1}\|_{\sup([-M,M]^D)} \to 0.$$

Then, for any $\varepsilon > 0$, there exists a small $\eta_0 > 0$ such that

$$\|\phi_{\eta_0} - f\|_{\sup([-M,M]^D)} < \varepsilon.$$

Let $g = \phi_{\eta_0}$ and the proof is finished by pointing out that

$$g \in \mathcal{NN}_{\sigma_L}(W, L, \mathbb{R}^D \to \mathbb{R}^{D'}).$$

$\square$

### E.4 Proof of Lemma E.3

The following proof basically follows (Jiao et al., 2025a).

*Proof of Lemma E.3.* According to the definition of FFN, $\boldsymbol{f}$ has the following form

$$\boldsymbol{f} = \boldsymbol{\mathcal{L}}_L \circ \cdots \circ \boldsymbol{\mathcal{L}}_0,$$

where each $\boldsymbol{\mathcal{L}}_\ell$ is given by $\boldsymbol{\mathcal{L}}_\ell(\boldsymbol{x}) := \boldsymbol{W}_\ell \boldsymbol{x}$ for $\ell = 0, 1 \cdots, L$ with $\boldsymbol{W}_\ell \in \mathbb{R}^{D_{\ell+1} \times D_\ell}$, $\boldsymbol{b}_\ell \in \mathbb{R}^{D_{\ell+1}}$, and $D_0 = D, D_1, \cdots, D_L \in \mathbb{N}^+$, and $D_{L+1} = D'$. Without loss of generality, we assume that $D_1, \cdots, D_L = W$, which can be achieved by simply zero-padding these weight matrices.

In the following, We consider two cases: **(1):** $D \leq W$, **(2):** $D > W$.

**Case 1:** $D \leq W$. Define $\widehat{\boldsymbol{\mathcal{L}}}_1$ as

$$\widehat{\boldsymbol{\mathcal{L}}}_1(\widehat{\boldsymbol{X}}) = \widehat{\boldsymbol{X}} + \widehat{\boldsymbol{W}}_1^{(2)} \sigma_L \left( \widehat{\boldsymbol{W}}_1^{(1)} \widehat{\boldsymbol{X}} \right),$$

where

$$\widehat{\boldsymbol{W}}_1^{(1)} = \begin{pmatrix} \boldsymbol{W}_0 & \boldsymbol{0}_{W \times (W-D)} \\ \boldsymbol{I}_D & \boldsymbol{0}_{D \times (W-D)} \\ -\boldsymbol{I}_D & \boldsymbol{0}_{D \times (W-D)} \end{pmatrix} \in \mathbb{R}^{(W+2D) \times W},$$

$$\widehat{\boldsymbol{W}}_1^{(2)} = \begin{pmatrix} \boldsymbol{I}_D & \boldsymbol{0} & -\boldsymbol{I}_D & \boldsymbol{I}_D \\ \boldsymbol{0} & \boldsymbol{I}_{W-D} & \boldsymbol{0} & \boldsymbol{0} \end{pmatrix} \in \mathbb{R}^{W \times (W+2D)}.$$

Direct computation yields

$$\widehat{\boldsymbol{\mathcal{L}}}_1(\widehat{\boldsymbol{X}}) = \begin{pmatrix} \boldsymbol{X} \\ \boldsymbol{0} \end{pmatrix} + \begin{pmatrix} \boldsymbol{I}_D & \boldsymbol{0} & -\boldsymbol{I}_D & \boldsymbol{I}_D \\ \boldsymbol{0} & \boldsymbol{I}_{W-D} & \boldsymbol{0} & \boldsymbol{0} \end{pmatrix} \sigma_L \left[ \begin{pmatrix} \boldsymbol{W}_0 & \boldsymbol{0}_{W \times (W-D)} \\ \boldsymbol{I}_D & \boldsymbol{0}_{D \times (W-D)} \\ -\boldsymbol{I}_D & \boldsymbol{0}_{D \times (W-D)} \end{pmatrix} \begin{pmatrix} \boldsymbol{X} \\ \boldsymbol{0} \end{pmatrix} \right]$$

$$= \sigma_L(\boldsymbol{W}_0 \boldsymbol{X}) + \begin{pmatrix} \boldsymbol{X} - \sigma_L(\boldsymbol{X}) + \sigma_L(\boldsymbol{X}) \\ \boldsymbol{0}_{(W-D) \times N} \end{pmatrix}$$

$$= \sigma_L \left[ \boldsymbol{\mathcal{L}}_0(\boldsymbol{X}) \right]$$

The last equality comes from the fact that

$$\sigma_L(\boldsymbol{X}) - \sigma_L(-\boldsymbol{X}) = \boldsymbol{X}.$$

For $\ell = 2, \ldots L - 1$, we define $\widehat{\boldsymbol{\mathcal{L}}}_\ell(\boldsymbol{Z}) := \boldsymbol{Z} + \widehat{\boldsymbol{W}}_\ell^{(2)} \sigma_L \left( \widehat{\boldsymbol{W}}_1^{(1)} \boldsymbol{Z} \right)$ for any $\boldsymbol{Z} \in \mathbb{R}^{W \times N}$, where

$$\widehat{\boldsymbol{W}}_\ell^{(1)} = \begin{pmatrix} \boldsymbol{W}_{\ell-1} \\ \boldsymbol{I}_W \\ -\boldsymbol{I}_W \end{pmatrix} \in \mathbb{R}^{3W \times W},$$

$$\widehat{\boldsymbol{W}}_\ell^{(2)} = \left( \boldsymbol{I}_W, -\boldsymbol{I}_W, \boldsymbol{I}_W \right) \in \mathbb{R}^{W \times (3W)}.$$

It is direct to verify that

$$\widehat{\boldsymbol{\mathcal{L}}}_{L-1} \circ \cdots \circ \widehat{\boldsymbol{\mathcal{L}}}_2 \left( \sigma_L \left( \boldsymbol{\mathcal{L}}_0(\boldsymbol{X}) \right) \right) = \sigma_L \circ \boldsymbol{\mathcal{L}}_{L-2} \circ \cdots \circ \sigma_L \circ \boldsymbol{\mathcal{L}}_0(\boldsymbol{X}).$$

For last layer, we define

$$\widehat{\boldsymbol{W}}_L^{(1)} = \begin{pmatrix} \boldsymbol{W}_{L-1} \\ \boldsymbol{I}_W \\ -\boldsymbol{I}_W \end{pmatrix} \in \mathbb{R}^{3W \times W},$$

$$\widehat{\boldsymbol{W}}_L^{(2)} = \begin{pmatrix} \boldsymbol{W}_L & -\boldsymbol{I}_{D'} & \boldsymbol{0} & \boldsymbol{I}_{D'} & \boldsymbol{0} \\ \boldsymbol{0} & \boldsymbol{0} & -\boldsymbol{I}_{W-D'} & \boldsymbol{0} & \boldsymbol{I}_{W-D'} \end{pmatrix} \in \mathbb{R}^{W \times 3W}.$$

Let $\boldsymbol{Z} = \sigma_L \circ \mathcal{L}_{L-2} \circ \cdots \circ \sigma_L \circ \mathcal{L}_0(\boldsymbol{X})$. Through direct computation, we have

$$\widehat{\mathcal{L}}_L(\boldsymbol{Z}) = \boldsymbol{Z} + \begin{pmatrix} \boldsymbol{W}_L & -\boldsymbol{I}_{D'} & \boldsymbol{0} & \boldsymbol{I}_{D'} & \boldsymbol{0} \\ \boldsymbol{0} & \boldsymbol{0} & -\boldsymbol{I}_{W-D'} & \boldsymbol{0} & \boldsymbol{I}_{W-D'} \end{pmatrix} \sigma_L \left( \begin{pmatrix} \boldsymbol{W}_{L-1} \\ \boldsymbol{I}_W \\ -\boldsymbol{I}_W \end{pmatrix} \boldsymbol{Z} \right)$$

$$= \begin{pmatrix} \boldsymbol{W}_L \sigma_L (\boldsymbol{W}_{L-1} \boldsymbol{Z}) \\ \boldsymbol{0}_{(W-D') \times N} \end{pmatrix} + \boldsymbol{Z} - \sigma_L(-\boldsymbol{Z}) + \sigma_L(\boldsymbol{Z})$$

$$= \begin{pmatrix} \boldsymbol{f}(\boldsymbol{X}) \\ \boldsymbol{0} \end{pmatrix} \in \mathbb{R}^{W \times N}.$$

which finishes the proof of **Case 1**.

**Case 2:** $D > W$. It is clear that we can zero-pad the weight matrices such that

$$\boldsymbol{W}_0 \in \mathbb{R}^{D \times D},$$
$$\boldsymbol{W}_\ell \in \mathbb{R}^{D \times D}, \quad \text{for } \ell = 1, \cdots, L-1,$$
$$\boldsymbol{W}_L \in \mathbb{R}^{D' \times D}.$$

which does not effect the operation of $\boldsymbol{f}$. Then, **Case 2** is reduced to **Case 1**, which completes the proof. $\square$

### E.5 PROOF OF LEMMA E.4

The following proof is built upon the techniques in (Huben & Morris, 2023).

*Proof of Lemma E.4.* Since $\boldsymbol{W}_2 \boldsymbol{W}_1 \in \mathbb{R}^{D \times D}$, we have $\boldsymbol{W}_2 \boldsymbol{W}_1 = \sum_{i=1}^{D'} \boldsymbol{a}_i \boldsymbol{b}_i^\top$, where $\boldsymbol{a}_i \in \mathbb{R}^{D \times 1}$ is the $i$-th column of $\boldsymbol{W}_2$ and $\boldsymbol{b}_i^\top \in \mathbb{R}^{1 \times D}$ is the $i$-th row of $\boldsymbol{W}_1$. We define

$$\boldsymbol{W}_K^{(i)} = [0, \cdots, 0, -1] \in \mathbb{R}^{1 \times (D+1)},$$
$$\boldsymbol{W}_Q^{(i)} = [\boldsymbol{b}_i^\top, 0, \cdots, 0] \in \mathbb{R}^{1 \times (D+1)},$$
$$\boldsymbol{W}_V^{(i)} = [\boldsymbol{b}_i^\top, 0, \cdots, 0] \in \mathbb{R}^{1 \times (D+1)},$$
$$\boldsymbol{W}_O^{(i)} = [\boldsymbol{a}_i, 0 \cdots, 0]^\top \in \mathbb{R}^{(D+1) \times 1}.$$

Through direct computation, the following holds

$$\left( \boldsymbol{W}_K^{(i)} \widehat{\boldsymbol{X}} \right)^\top \left( \boldsymbol{W}_Q^{(i)} \widehat{\boldsymbol{X}} \right) = \begin{pmatrix} 0 & \cdots & 0 & 0 \\ \vdots & & \vdots & \vdots \\ 0 & \cdots & 0 & 0 \\ -\boldsymbol{b}_i^\top \boldsymbol{X} & & & 0 \end{pmatrix} \in \mathbb{R}^{(N+1) \times (N+1)}.$$

We define the positional encoding matrix as

$$\boldsymbol{R}^{(i)} = \begin{pmatrix} 0 & -\infty & \cdots & -\infty & -\infty \\ -\infty & \ddots & & \vdots & \vdots \\ \vdots & & \ddots & -\infty & -\infty \\ -\infty & \cdots & -\infty & 0 & -\infty \\ 0 & \cdots & 0 & 0 & 0 \end{pmatrix} \in \mathbb{R}^{(N+1) \times (N+1)},$$

where in the first $N+1$ columns, the interaction between $i$-th and $(N+1)$-th column, $i$-th and $i$-th token is allowed. In the last column, the interaction between $(N+1)$-th token and $(N+1)$-th token is allowed. Thus, we have

$$\sigma_S\left[\left(\boldsymbol{W}_K^{(i)}\widehat{\boldsymbol{X}}\right)^\top\left(\boldsymbol{W}_Q^{(i)}\widehat{\boldsymbol{X}}\right)\right]$$

$$=\begin{pmatrix} \frac{1}{1+\exp\left(-\boldsymbol{b}_i^\top\boldsymbol{X}_{:,1}\right)} & 0 & \cdots & \cdots & \cdots & 0 \\ 0 & \frac{1}{1+\exp\left(-\boldsymbol{b}_i^\top\boldsymbol{X}_{:,2}\right)} & 0 & \cdots & \cdots & 0 \\ \vdots & \vdots & \ddots & & \cdots & \vdots \\ 0 & 0 & \ddots & \frac{1}{1+\exp\left(-\boldsymbol{b}_i^\top\boldsymbol{X}_{:,N}\right)} & 0 \\ \frac{\exp\left(-\boldsymbol{b}_i^\top\boldsymbol{X}_{:,1}\right)}{1+\exp\left(-\boldsymbol{b}_i^\top\boldsymbol{X}_{:,1}\right)} & \frac{\exp\left(-\boldsymbol{b}_i^\top\boldsymbol{X}_{:,2}\right)}{1+\exp\left(-\boldsymbol{b}_i^\top\boldsymbol{X}_{:,2}\right)} & & \frac{\exp\left(-\boldsymbol{b}_i^\top\boldsymbol{X}_{:,N}\right)}{1+\exp\left(-\boldsymbol{b}_i^\top\boldsymbol{X}_{:,N}\right)} & 1 \end{pmatrix}.$$

Taking $\boldsymbol{W}_V$ into consideration, we have

$$\boldsymbol{W}_V^{(i)}\widehat{\boldsymbol{X}}\sigma_S\left[\left(\boldsymbol{W}_K^{(i)}\widehat{\boldsymbol{X}}\right)^\top\left(\boldsymbol{W}_Q^{(i)}\widehat{\boldsymbol{X}}\right)\right]$$

$$=\left(\frac{\boldsymbol{b}_i^\top\boldsymbol{X}_{:,1}}{1+\exp\left(-\boldsymbol{b}_i^\top\boldsymbol{X}_{:,1}\right)} \quad \frac{\boldsymbol{b}_i^\top\boldsymbol{X}_{:,2}}{1+\exp\left(-\boldsymbol{b}_i^\top\boldsymbol{X}_{:,2}\right)} \quad \cdots \quad \frac{\boldsymbol{b}_i^\top\boldsymbol{X}_{:,N}}{1+\exp\left(-\boldsymbol{b}_i^\top\boldsymbol{X}_{:,N}\right)} \quad 0\right)$$

$$=\left(\sigma_L(\boldsymbol{b}_i^\top\boldsymbol{X}_{:,1}),\sigma_L(\boldsymbol{b}_i^\top\boldsymbol{X}_{:,2}),\cdots,\sigma_L(\boldsymbol{b}_i^\top\boldsymbol{X}_{:,N}),0\right)\in\mathbb{R}^{1\times(N+1)}.$$

Finally, $\boldsymbol{W}_O^{(i)}$ recovers the matrix to the original size

$$\boldsymbol{W}_O^{(i)}\boldsymbol{W}_V^{(i)}\widehat{\boldsymbol{X}}\sigma_S\left[\left(\boldsymbol{W}_K^{(i)}\widehat{\boldsymbol{X}}\right)^\top\left(\boldsymbol{W}_Q^{(i)}\widehat{\boldsymbol{X}}\right)\right]$$

$$=\begin{pmatrix} \boldsymbol{a}_i\sigma_L(\boldsymbol{b}_i^\top\boldsymbol{X}),\cdots,\boldsymbol{a}_i\sigma_L(\boldsymbol{b}_i^\top\boldsymbol{X}) & \boldsymbol{0} \\ \boldsymbol{0} & 0 \end{pmatrix}\in\mathbb{R}^{(D+1)\times(N+1)}.$$

Since $\boldsymbol{W}_2\boldsymbol{W}_1=\sum_{i=1}^{D'}\boldsymbol{a}_i\boldsymbol{b}_i^\top$ and $\sigma_L$ is element-wise, it is straightforward to have by incorporating the skip connection

$$\widehat{\boldsymbol{X}}+\sum_{i=1}^{D'}\boldsymbol{W}_O^{(i)}\boldsymbol{W}_V^{(i)}\widehat{\boldsymbol{X}}\sigma_S\left[\left(\boldsymbol{W}_K^{(i)}\widehat{\boldsymbol{X}}\right)^\top\left(\boldsymbol{W}_Q^{(i)}\widehat{\boldsymbol{X}}\right)\right]$$

$$=\begin{pmatrix} \boldsymbol{W}_2\sigma_L(\boldsymbol{W}_1\boldsymbol{X})+\boldsymbol{X} & \boldsymbol{0} \\ \boldsymbol{0} & 1 \end{pmatrix}\in\mathbb{R}^{(D+1)\times(N+1)}.$$

This part is finished by constructing $\boldsymbol{\mathcal{F}}_{SA}$ with $\{\boldsymbol{W}_O^{(i)},\boldsymbol{W}_V^{(i)},\boldsymbol{W}_K^{(i)},\boldsymbol{W}_Q^{(i)}\}_{i\in[D']}$. $\qquad\square$

### E.6 PROOF OF COROLLARY 4.1

We only need to modify Lemma E.4 in the proof of Theorem 4.1 and keep other steps unchanged.

*Proof of Corollary 4.1.* For any vector $\boldsymbol{v}\in\mathbb{R}^{\max\{W,D\}+2}$ with the last element $\boldsymbol{v}_{-1}\neq0$, We redefine the following matrices in the proof of Lemma E.4

$$\boldsymbol{W}_K^{(i)}=[0,\cdots,0,-1/\boldsymbol{v}_{-1}]\in\mathbb{R}^{1\times(D+1)},$$
$$\boldsymbol{W}_Q^{(i)}=[\boldsymbol{b}_i^\top,0,\cdots,0]\in\mathbb{R}^{1\times(D+1)},$$
$$\boldsymbol{W}_V^{(i)}=[\boldsymbol{b}_i^\top,0,\cdots,0]\in\mathbb{R}^{1\times(D+1)},$$
$$\boldsymbol{W}_O^{(i)}=[\boldsymbol{a}_i,0\cdots,0]^\top\in\mathbb{R}^{(D+1)\times1}.$$

Through direct computation, the following holds

$$\left(\boldsymbol{W}_K^{(i)}\widehat{\boldsymbol{X}}\right)^\top \left(\boldsymbol{W}_Q^{(i)}\widehat{\boldsymbol{X}}\right) = \begin{pmatrix} 0 & \cdots & 0 & 0 \\ \vdots & & \vdots & \vdots \\ 0 & \cdots & 0 & 0 \\ -\boldsymbol{b}_i^\top \boldsymbol{X} & & [-\boldsymbol{b}_i^\top, 0]\boldsymbol{v} \end{pmatrix} \in \mathbb{R}^{(N+1)\times(N+1)}.$$

Similarly, we define the mask matrix as

$$\boldsymbol{R}^{(i)} = \begin{pmatrix} 0 & -\infty & \cdots & -\infty & -\infty \\ -\infty & \ddots & & \vdots & \vdots \\ \vdots & & \ddots & -\infty & -\infty \\ -\infty & \cdots & -\infty & 0 & -\infty \\ 0 & \cdots & 0 & 0 & -\infty \end{pmatrix} \in \mathbb{R}^{(N+1)\times(N+1)},$$

where in the first $N + 1$ columns, the interaction between $i$-th and $(N + 1)$-th column, $i$-th and $i$-th token is allowed. In the last column, the interaction between $(N + 1)$-th token and $(N + 1)$-th token is allowed. Thus, we have

$$\sigma_S\left[\left(\boldsymbol{W}_K^{(i)}\widehat{\boldsymbol{X}}\right)^\top \left(\boldsymbol{W}_Q^{(i)}\widehat{\boldsymbol{X}}\right)\right]$$

$$= \begin{pmatrix} \frac{1}{1+\exp\left(-\boldsymbol{b}_i^\top \boldsymbol{X}_{:,1}\right)} & 0 & \cdots & \cdots & \cdots & 0 \\ 0 & \frac{1}{1+\exp\left(-\boldsymbol{b}_i^\top \boldsymbol{X}_{:,2}\right)} & 0 & \cdots & \cdots & 0 \\ \vdots & \vdots & \ddots & & \cdots & \vdots \\ 0 & 0 & \ddots & \frac{1}{1+\exp\left(-\boldsymbol{b}_i^\top \boldsymbol{X}_{:,N}\right)} & 0 \\ \frac{\exp\left(-\boldsymbol{b}_i^\top \boldsymbol{X}_{:,1}\right)}{1+\exp\left(-\boldsymbol{b}_i^\top \boldsymbol{X}_{:,1}\right)} & \frac{\exp\left(-\boldsymbol{b}_i^\top \boldsymbol{X}_{:,2}\right)}{1+\exp\left(-\boldsymbol{b}_i^\top \boldsymbol{X}_{:,2}\right)} & & \frac{\exp\left(-\boldsymbol{b}_i^\top \boldsymbol{X}_{:,N}\right)}{1+\exp\left(-\boldsymbol{b}_i^\top \boldsymbol{X}_{:,N}\right)} & 0 \end{pmatrix}.$$

Taking $\boldsymbol{W}_V^{(i)}$ into consideration, we have

$$\boldsymbol{W}_V^{(i)}\widehat{\boldsymbol{X}}\sigma_S\left[\left(\boldsymbol{W}_K^{(i)}\widehat{\boldsymbol{X}}\right)^\top \left(\boldsymbol{W}_Q^{(i)}\widehat{\boldsymbol{X}}\right)\right]$$

$$= \left(\frac{\boldsymbol{b}_i^\top \boldsymbol{X}_{:,1}}{1+\exp\left(-\boldsymbol{b}_i^\top \boldsymbol{X}_{:,1}\right)} \quad \frac{\boldsymbol{b}_i^\top \boldsymbol{X}_{:,2}}{1+\exp\left(-\boldsymbol{b}_i^\top \boldsymbol{X}_{:,2}\right)} \quad \cdots \quad \frac{\boldsymbol{b}_i^\top \boldsymbol{X}_{:,N}}{1+\exp\left(-\boldsymbol{b}_i^\top \boldsymbol{X}_{:,N}\right)} \quad 0\right)$$

$$= \left(\sigma_L(\boldsymbol{b}_i^\top \boldsymbol{X}_{:,1}), \sigma_L(\boldsymbol{b}_i^\top \boldsymbol{X}_{:,2}), \cdots, \sigma_L(\boldsymbol{b}_i^\top \boldsymbol{X}_{:,N}), 0\right) \in \mathbb{R}^{1\times(N+1)}.$$

Finally, $\boldsymbol{W}_O^{(i)}$ recovers the matrix to the original size

$$\boldsymbol{W}_O^{(i)}\boldsymbol{W}_V^{(i)}\widehat{\boldsymbol{X}}\sigma_S\left[\left(\boldsymbol{W}_K^{(i)}\widehat{\boldsymbol{X}}\right)^\top \left(\boldsymbol{W}_Q^{(i)}\widehat{\boldsymbol{X}}\right)\right]$$

$$= \begin{pmatrix} \boldsymbol{a}_i\sigma_L(\boldsymbol{b}_i^\top \boldsymbol{X}), \cdots, \boldsymbol{a}_i\sigma_L(\boldsymbol{b}_i^\top \boldsymbol{X}) & \boldsymbol{0} \\ \boldsymbol{0} & 0 \end{pmatrix} \in \mathbb{R}^{(D+1)\times(N+1)}.$$

Since $\boldsymbol{W}_2\boldsymbol{W}_1 = \sum_{i=1}^{D'} \boldsymbol{a}_i\boldsymbol{b}_i^\top$ and $\sigma_L$ is element-wise, it is straightforward to have by incorporating the skip connection

$$\widehat{\boldsymbol{X}} + \sum_{i=1}^{D'} \boldsymbol{W}_O^{(i)}\boldsymbol{W}_V^{(i)}\widehat{\boldsymbol{X}}\sigma_S\left[\left(\boldsymbol{W}_K^{(i)}\widehat{\boldsymbol{X}}\right)^\top \left(\boldsymbol{W}_Q^{(i)}\widehat{\boldsymbol{X}}\right)\right]$$

$$= \begin{pmatrix} \boldsymbol{W}_2\sigma_L(\boldsymbol{W}_1\boldsymbol{X}) + \boldsymbol{X} & \boldsymbol{0} \\ \boldsymbol{0} & 1 \end{pmatrix} \in \mathbb{R}^{(D+1)\times(N+1)}.$$

The proof is finished by constructing $\mathcal{F}_{SA}$ with $\{\boldsymbol{W}_O^{(i)}, \boldsymbol{W}_V^{(i)}, \boldsymbol{W}_K^{(i)}, \boldsymbol{W}_Q^{(i)}\}_{i\in[D']}$. $\qquad\square$

# F    PROOF OF SECTION 5

## F.1    PROOF OF PROPOSITION 5.1

**Proposition F.1** (Restatement of Proposition 5.1). *For any input-label pairs* $(\boldsymbol{X}^{(1)}, y^{(1)}), \cdots, (\boldsymbol{X}^{(n)}, y^{(n)})$ *satisfying Assumption 3.1, there exists a self-attention layer* $\boldsymbol{\mathcal{F}}_{SA} \in \mathcal{T}(D, 1, 1, 1)$ *such that*

$$\boldsymbol{\mathcal{F}}_{SA}(\boldsymbol{X}^{(i)})_{:,m_i} \neq \boldsymbol{\mathcal{F}}_{SA}(\boldsymbol{X}^{(j)})_{:,m_j} \quad \text{for any } i \neq j \in [n].$$

*Proof of Proposition 5.1.* Let $\mathcal{X} := \{\boldsymbol{X}^{(i)}_{:,j} \mid \text{for any } i \in [n], j \in [N]\}$. According to Lemma H.2, for any $\delta > 0$, there exists $\boldsymbol{W}_k, \boldsymbol{W}_Q \in \mathbb{R}^{1 \times D}$ such that

$$\left| (\boldsymbol{W}_K \boldsymbol{x}_a)^\top (\boldsymbol{W}_Q \boldsymbol{x}_c) - (\boldsymbol{W}_K \boldsymbol{x}_b)^\top (\boldsymbol{W}_Q \boldsymbol{x}_c) \right| > \delta \qquad \text{(F.1)}$$

for any $\boldsymbol{x}_a, \boldsymbol{x}_b, \boldsymbol{x}_c \in \mathcal{X}$ with $\boldsymbol{x}_a \neq \boldsymbol{x}_b$. $\boldsymbol{W}_K, \boldsymbol{W}_Q$ have the following form

$$\boldsymbol{W}_K = \boldsymbol{u} \boldsymbol{v}^\top, \boldsymbol{W}_Q = \boldsymbol{u}' \boldsymbol{v}^\top$$

where $\boldsymbol{u}, \boldsymbol{u}' \in \mathbb{R}$ and $\boldsymbol{v} \in \mathbb{R}^{D \times 1}$.

Define $\boldsymbol{a}^{(i)}$ and $\boldsymbol{a}^{(j)}$ by

$$\boldsymbol{a}^{(i)} = \left( \boldsymbol{W}_K \boldsymbol{X}^{(i)} \right)^\top \left( \boldsymbol{W}^{(Q)} \boldsymbol{X}^{(i)}_{:,m_i} \right),$$

$$\boldsymbol{a}^{(j)} = \left( \boldsymbol{W}_K \boldsymbol{X}^{(j)} \right)^\top \left( \boldsymbol{W}^{(Q)} \boldsymbol{X}^{(j)}_{:,m_j} \right).$$

Then, equation F.1 shows that

$$\| \boldsymbol{a}^{(i)} - \boldsymbol{a}^{(j)} \| > \delta.$$

Since there are no duplicated tokens in $\boldsymbol{X}^{(i)}$ for any $i \in [N]$, it follows from Lemma H.3 that

$$\left( \boldsymbol{a}^{(i)} \right)^\top \sigma_S \left[ \boldsymbol{a}^{(i)} \right] \neq \left( \boldsymbol{a}^{(j)} \right)^\top \sigma_S \left[ \boldsymbol{a}^{(j)} \right]$$

by letting $\delta > 2 \log D + 3$. This implies that there exists $\delta' > 0$ such that

$$\begin{aligned}
\delta' &< \left| \left( \boldsymbol{a}^{(i)} \right)^\top \sigma_S \left[ \boldsymbol{a}^{(i)} \right] - \left( \boldsymbol{a}^{(j)} \right)^\top \sigma_S \left[ \boldsymbol{a}^{(j)} \right] \right| \\
&= \left| \left( \boldsymbol{X}^{(i)}_{:,m_i} \right)^\top (\boldsymbol{W}_Q)^\top \boldsymbol{W}_K \left( \boldsymbol{X}^{(i)} \sigma_S \left[ \boldsymbol{a}^{(i)} \right] - \boldsymbol{X}^{(j)} \sigma_S \left[ \boldsymbol{a}^{(j)} \right] \right) \right| \\
&= \left| \left( \boldsymbol{X}^{(i)}_{:,m_i} \right) \boldsymbol{v} \boldsymbol{u}' \boldsymbol{u} \boldsymbol{v}^\top \left( \boldsymbol{X}^{(i)} \sigma_S \left[ \boldsymbol{a}^{(i)} \right] - \boldsymbol{X}^{(j)} \sigma_S \left[ \boldsymbol{a}^{(j)} \right] \right) \right| \\
&= \left| \boldsymbol{v}^\top \boldsymbol{X}^{(i)}_{:,m_i} \right| \cdot |\boldsymbol{u} \boldsymbol{u}'| \cdot \left| \left( \boldsymbol{v}^\top \boldsymbol{X}^{(i)} \sigma_S \left[ \boldsymbol{a}^{(i)} \right] - \boldsymbol{v}^\top \boldsymbol{X}^{(j)} \sigma_S \left[ \boldsymbol{a}^{(j)} \right] \right) \right|,
\end{aligned}$$

which means that

$$\boldsymbol{v}^\top \boldsymbol{X}^{(i)} \sigma_S \left[ \boldsymbol{a}^{(i)} \right] \neq \boldsymbol{v}^\top \boldsymbol{X}^{(j)} \sigma_S \left[ \boldsymbol{a}^{(j)} \right].$$

We construct $\boldsymbol{\mathcal{F}}$ by

$$\begin{aligned}
\boldsymbol{W}_K &= \boldsymbol{u} \boldsymbol{v}^\top, \\
\boldsymbol{W}_Q &= \boldsymbol{u}' \boldsymbol{v}^\top, \\
\boldsymbol{W}_V &= \boldsymbol{v}^\top, \\
\boldsymbol{W}_O &= \boldsymbol{u}'',
\end{aligned}$$

where $\boldsymbol{u}, \boldsymbol{u}', \boldsymbol{v}$ are defined in Lemma H.2, $\boldsymbol{u}'' \in \mathbb{R}^{D \times 1}$ is an arbitrary nonzero vector satisfying $\boldsymbol{u}'' \boldsymbol{v}^\top \neq \boldsymbol{0}$.

$$
\left\| \boldsymbol{\mathcal{F}}_{SA}(\boldsymbol{X}^{(i)})_{:,m_i} - \boldsymbol{\mathcal{F}}_{SA}(\boldsymbol{X}^{(j)})_{:,m_j} \right\|
$$
$$
= \left\| \boldsymbol{W}_O \left( \boldsymbol{W}_V \boldsymbol{X}^{(i)} \right) \sigma_S \left[ \boldsymbol{a}^{(i)} \right] - \boldsymbol{W}_O \left( \boldsymbol{W}_V \boldsymbol{X}^{(j)} \right) \sigma_S \left[ \boldsymbol{a}^{(j)} \right] \right\|
$$
$$
= \| \boldsymbol{u}'' \| \cdot \left| \left( \boldsymbol{v}^\top \boldsymbol{X}^{(i)} \sigma_S \left[ \boldsymbol{a}^{(i)} \right] \right) - \left( \boldsymbol{v}^\top \boldsymbol{X}^{(j)} \sigma_S \left[ \boldsymbol{a}^{(j)} \right] \right) \right|
$$
$$
\neq \boldsymbol{0}.
$$

The proof is completed by pointing out that

$$
\boldsymbol{\mathcal{F}}_{SA} \in \mathcal{T}(D, 1, 1, 1).
$$

$\square$

### F.2 PROOF OF PROPOSITION 5.2

**Proposition F.2** (Restatement of Proposition 5.2)**.** *For any input-label pairs $(\boldsymbol{X}^{(1)}, y^{(1)}), \cdots, (\boldsymbol{X}^{(n)}, y^{(n)})$ satisfying Assumption 3.1, there exists a self-attention layer $\boldsymbol{\mathcal{F}}_{SA}^{\boldsymbol{A}} \in \mathcal{T}(D, 1, 1, 1)$ with a fixed attention pattern $\boldsymbol{A}$ such that*

$$
\boldsymbol{\mathcal{F}}_{SA}^{\boldsymbol{A}}(\boldsymbol{X}^{(i)})_{:,m_i} \neq \boldsymbol{\mathcal{F}}_{SA}^{\boldsymbol{A}}(\boldsymbol{X}^{(j)})_{:,m_j} \quad \text{for any } i \neq j \in [n].
$$

*Proof of Proposition 5.2.* Without loss of generality, we assume that $m_1, \cdots, m_n$ are arranged from low to high. For each $i = 1, \cdots, n$ we define

$$
\mathcal{P}_i = \{ j : j \in \mathbb{Z}_{>0}, j < i, m_j = m_i \}.
$$

For any $\ell = 1, \cdots, n$, we prove by induction that there exists a set of vectors $\{ \boldsymbol{a}_{m_1}, \cdots, \boldsymbol{a}_{m_\ell} \} \subset \mathbb{R}^N$ such that

$$
\boldsymbol{X}^{(i)} \boldsymbol{a}_{m_i} \neq \boldsymbol{X}^{(j)} \boldsymbol{a}_{m_j} \quad \text{for any } i \neq j \in [\ell]. \tag{F.2}
$$

It is clear that when $\ell = 1$, Equation.F.2 is naturally satisfied. Next, supposing Equantion.F.2 holds for $\ell = m \geq 2$, we aim to prove that this is also holds for $\ell = m + 1$.

We consider the following set of vectors

$$
\mathcal{B}_{m+1} := \bigcup_{i \in \mathcal{P}_{m+1}} \ker \left( \boldsymbol{X}^{(i)} - \boldsymbol{X}^{(m+1)} \right).
$$

Each $\ker \left( \boldsymbol{X}^{(i)} - \boldsymbol{X}^{(m+1)} \right)$ is defined by

$$
\ker \left( \boldsymbol{X}^{(i)} - \boldsymbol{X}^{(m+1)} \right) := \left\{ \boldsymbol{x} \in \mathbb{R}^N : \left( \boldsymbol{X}^{(i)} - \boldsymbol{X}^{(m+1)} \right) \boldsymbol{x} = \boldsymbol{0} \right\}.
$$

According to Assumption 3.1, $\boldsymbol{X}^{(i)} \neq \boldsymbol{X}^{(j)}$ for any $i \neq j \in [n]$, meaning that

$$
\ker \left( \boldsymbol{X}^{(i)} - \boldsymbol{X}^{(m+1)} \right) \subsetneq \mathbb{R}^N \quad \text{for any } i \in \mathcal{P}_{m+1}.
$$

Furthermore, for each $i = 1, \cdots, m - |\mathcal{P}_{m+1}|$, define

$$
\mathcal{Q}_i := \{ \boldsymbol{a} \in \mathbb{R}^N : \boldsymbol{X}^{(i)} \boldsymbol{a}_{m_i} = \boldsymbol{X}^{(m+1)} \boldsymbol{a} \}.
$$

For any $i = 1, \cdots, m - |\mathcal{P}_{m+1}|$ and any $\boldsymbol{a} \in \mathcal{Q}_i$, we know that

$$
\boldsymbol{X}^{(i)} \boldsymbol{a}_{m_i} = \boldsymbol{X}^{(m+1)} \boldsymbol{a}.
$$

Since $\boldsymbol{X}^{(m+1)} \neq \boldsymbol{0}$, we know that $\mathcal{Q}_i \subsetneq \mathbb{R}^N$.

It is obvious that $\bigcup_{i=1,\cdots,m-|\mathcal{P}_{m+1}|} \mathcal{Q}_i \bigcup \mathcal{B}_{m+1}$ is a finite union of proper subspaces in $\mathbb{R}^N$, which has empty interior. Since $\mathbb{R}_{>0}^N$ has non empty interior, we know that there exists $\boldsymbol{a}_{m+1} \in \mathbb{R}^N$ such that

$$\boldsymbol{a}_{m+1} \in \mathbb{R}_{>0}^N,$$

$$\boldsymbol{a}_{m+1} \notin \bigcup_{i=1,\cdots,m-|\mathcal{P}_{m+1}|} \mathcal{Q}_i \bigcup \mathcal{B}_{m+1},$$

meaning that

$$\boldsymbol{X}^{(i)}\boldsymbol{a}_{m_i} \neq \boldsymbol{X}^{(m+1)}\boldsymbol{a}_{m+1} \quad \text{for any } i \in \mathcal{P}_{m+1},$$

$$\boldsymbol{X}^{(i)}\boldsymbol{a}_{m_i} \neq \boldsymbol{X}^{(m+1)}\boldsymbol{a}_{m+1} \quad \text{for any } i = 1, \cdots, m - |\mathcal{P}_{m+1}|.$$

Since we assume that Equation.F.2 holds for $\ell = m$, we have

$$\boldsymbol{X}^{(i)}\boldsymbol{a}_{:,m_i} \neq \boldsymbol{X}^{(j)}\boldsymbol{a}_{:,m_j} \quad \text{for any } i \neq j \in [\ell],$$

which completes the induction step. Up to now, we have proved that we can find proper $\{\boldsymbol{a}_{m_i}\}_{i\in[n]}$ such that

$$\boldsymbol{X}^{(i)}\boldsymbol{a}_{m_i} \neq \boldsymbol{X}^{(j)}\boldsymbol{a}_{m_j} \quad \text{for any } i \neq j \in [n].$$

By applying Lemma H.1 to $\boldsymbol{X}^{(1)}\boldsymbol{a}_{m_1}, \cdots, \boldsymbol{X}^{(n)}\boldsymbol{a}_{m_n}$, there exists a vector $\boldsymbol{v} \in \mathbb{R}^D$ such that $\boldsymbol{v}^\top \boldsymbol{X}^{(i)}\boldsymbol{a}_{m_i}$ are pair-wise distinct. Let $\boldsymbol{W}_V = \boldsymbol{v}^\top$ and $\boldsymbol{W}_O$ be an arbitrary non-zero vector in $\mathbb{R}^D$. We define

$$\boldsymbol{\mathcal{F}}_{SA}^{\boldsymbol{A}} := \boldsymbol{X} + \boldsymbol{W}_O \boldsymbol{W}_V \boldsymbol{X} \boldsymbol{A},$$

where $\boldsymbol{A}_{:,m_i} = \boldsymbol{a}_{m_i}$ and other columns are arbitrarily chosen from $\mathbb{R}_{>0}^N$. It is straightforward to verify that

$$|\boldsymbol{\mathcal{F}}_{SA}^{\boldsymbol{A}}(\boldsymbol{X}^{(i)})_{:,m_i} - \boldsymbol{\mathcal{F}}_{SA}^{\boldsymbol{A}}(\boldsymbol{X}^{(j)})_{:,m_j}|$$

$$= \left| \boldsymbol{X}_{:,m_i} + \boldsymbol{W}_O \boldsymbol{W}_V \boldsymbol{X} \boldsymbol{A}_{:,m_i} - \boldsymbol{X}_{:,m_j} - \boldsymbol{W}_O \boldsymbol{W}_V \boldsymbol{X} \boldsymbol{A}_{:,m_j} \right|$$

$$= \boldsymbol{W}_O \boldsymbol{v}^\top \left( \boldsymbol{X}^{(i)}\boldsymbol{a}_{m_i} - \boldsymbol{X}^{(j)}\boldsymbol{a}_{m_j} \right)$$

$$\neq \boldsymbol{0},$$

which completes the proof. $\qquad\square$

### F.3 Proof of Proposition 5.3

**Proposition F.3** (Restatement of Proposition 5.3). *For any input-label pairs* $(\boldsymbol{X}^{(1)}, y^{(1)}), \cdots, (\boldsymbol{X}^{(n)}, y^{(n)})$ *satisfying Assumption 3.1, and any attention pattern* $\boldsymbol{A} \in \mathbb{R}^{N \times N}$ *with* $\boldsymbol{A}_{i,j} > 0$ *for any* $i, j \in [N]$, *there exists a Transformer* $\boldsymbol{\mathcal{F}} = \boldsymbol{\mathcal{E}}_{out} \circ \boldsymbol{\mathcal{F}}_{SA}^{\boldsymbol{A}} \circ \boldsymbol{\mathcal{F}}_{FF} \circ \boldsymbol{\mathcal{E}}_{in} \in \mathcal{T}(\max\{3(n-1)N, D\}, 1, 1, 3(n-1)N, 1)$ *such that*

$$\boldsymbol{\mathcal{F}}(\boldsymbol{X}^{(i)})_{:,m_i} \neq \boldsymbol{\mathcal{F}}(\boldsymbol{X}^{(j)})_{:,m_j} \quad \text{for any } i \neq j \in [n].$$

*Proof of Proposition 5.3.* Let $S = \{\boldsymbol{X}_{:,j}^{(i)} \mid i \in [n], j \in [N]\}$, which contains all tokens in $\boldsymbol{X}^{(i)}$. It is clear that $|S| \leq nN$. Let $g : S \to [|S|] = \{1, 2, \cdots, |S|\}$ be an arbitrary bijective function. For each $\boldsymbol{X}^{(i)}$ with $i \in [n]$, we define

$$\boldsymbol{m}^{(i)} = \sum_{j=1}^N \boldsymbol{A}_{j,m_i} \boldsymbol{e}_{g(\boldsymbol{X}_{:,j}^{(i)})} \in \mathbb{N}^{|S|},$$

where $\boldsymbol{e}_{g(\boldsymbol{X}_{:,j}^{(i)})}$ is a one-hot vector with 1 in the $g(\boldsymbol{X}_{:,j}^{(i)})$-position. Since there are no repeated tokens in $\boldsymbol{X}^{(i)}$ and for any $i \neq j \in [n]$ $\boldsymbol{X}^{(i)}$ and $\boldsymbol{X}^{(j)}$ are not permutation equivalent, meaning that

$$\left\| \boldsymbol{m}^{(i)} - \boldsymbol{m}^{(j)} \right\|^2 > 0,$$

for any $i \neq j \in [n]$. By applying Lemma H.1 to $\boldsymbol{m}^{(1)}, \cdots, \boldsymbol{m}^{(n)}$, there exists a vector $\boldsymbol{v} \in \mathbb{R}^{|S|}$ such that

$$\frac{1}{n^2}\sqrt{\frac{8}{\pi|S|}}\left\|\boldsymbol{m}^{(i)} - \boldsymbol{m}^{(j)}\right\| \leq \left|\boldsymbol{v}^\top\left(\boldsymbol{m}^{(i)} - \boldsymbol{m}^{(j)}\right)\right| \leq \left\|\boldsymbol{m}^{(i)} - \boldsymbol{m}^{(j)}\right\|$$

holds for any $i, j \in [n]$. Let $\boldsymbol{h}$ be the function $\boldsymbol{h} : S \to \mathbb{R}$, $\boldsymbol{x} \mapsto \boldsymbol{v}_{g(\boldsymbol{x})}$. Notice that

$$\sum_{j=1}^{N} \boldsymbol{A}_{j,m_i} \boldsymbol{h}(\boldsymbol{X}_{:,j}^{(i)}) = \sum_{j=1}^{N} \boldsymbol{A}_{j,m_i} \boldsymbol{v}_{g(\boldsymbol{X}_{:,j}^{(i)})} = \boldsymbol{v}^\top \boldsymbol{m}^{(i)}$$

holds for any $i \in [n]$. Then, for any $k \neq l \in [n]$, we have

$$\left|\sum_{j=1}^{N} \boldsymbol{A}_{j,m_k} \boldsymbol{h}(\boldsymbol{X}_{:,j}^{(k)}) - \sum_{j=1}^{N} \boldsymbol{A}_{j,m_l} \boldsymbol{h}(\boldsymbol{X}_{:,j}^{(l)})\right|$$

$$= \left|\boldsymbol{v}^\top\left(\boldsymbol{m}^{(k)} - \boldsymbol{m}^{(l)}\right)\right|$$

$$\geq \frac{1}{n^2}\sqrt{\frac{8}{\pi|S|}}\left\|\boldsymbol{m}^{(k)} - \boldsymbol{m}^{(l)}\right\| > 0.$$

In the following, we use one feed-forward layer to implement $\boldsymbol{f}(\cdot)$. By applying Lemma H.4 to the set $\{(\boldsymbol{X}_{:,j}^{(i)})\}_{i\in[n],j\in[N]}$, there exists a $\sigma_R$-activated FFN $\boldsymbol{f} \in \mathcal{NN}_{\sigma_R}(3(n-1)N, 1, \mathbb{R}^D \to \mathbb{R})$ such that

$$\boldsymbol{f}(\boldsymbol{X}_{:,j}^{(i)}) = \boldsymbol{h}(\boldsymbol{X}_{:,j}^{(i)}).$$

Moreover, according to Lemma E.3, there exists a residual FFN $\hat{\boldsymbol{f}} \in \mathcal{NN}_{\sigma_R}^{Res}(\max\{3(n-1)N, D\}, 1, \mathbb{R}^{\max\{3(n-1)N,D\}} \to \mathbb{R}^{\max\{3(n-1)N,D\}})$ such that

$$\hat{\boldsymbol{f}}(\widehat{\boldsymbol{X}}) = \begin{pmatrix} \boldsymbol{f}(\boldsymbol{X}) \\ \boldsymbol{0} \end{pmatrix} = \begin{pmatrix} \boldsymbol{h}(\boldsymbol{X}) \\ \boldsymbol{0} \end{pmatrix},$$

where $\widehat{\boldsymbol{X}} = \begin{pmatrix} \boldsymbol{X} \\ \boldsymbol{0}_{(3(n-1)N-D)^+ \times N} \end{pmatrix}$. Define $\mathcal{F}_{SA}$ by

$$\mathcal{F}_{SA}(\boldsymbol{X}) := \boldsymbol{X} + \boldsymbol{W}_O \boldsymbol{W}_V \boldsymbol{X} \boldsymbol{A},$$

where

$$\boldsymbol{W}_O = \begin{pmatrix} 0 & 1 & 0 & \cdots & 0 \end{pmatrix}^\top \in \mathbb{R}^{\max\{3(n-1)N,D\} \times 1},$$

$$\boldsymbol{W}_V = \begin{pmatrix} 1 & 1 & 1 & \cdots & 1 \end{pmatrix} \in \mathbb{R}^{1 \times \max\{3(n-1)N,D\}}.$$

It is straightforward to verify that

$$\mathcal{F}_{SA}\left(\hat{\boldsymbol{f}}(\widehat{\boldsymbol{X}})\right) = \begin{pmatrix} \boldsymbol{h}(\boldsymbol{X}) \\ \boldsymbol{h}(\boldsymbol{X})\boldsymbol{A} \\ \boldsymbol{0} \end{pmatrix} = \begin{pmatrix} \boldsymbol{h}(\boldsymbol{X}_{:,1}) & \cdots & \boldsymbol{h}(\boldsymbol{X}_{:,N}) \\ \sum_{j=1}^{N}\boldsymbol{A}_{j,1}\boldsymbol{h}(\boldsymbol{X}_{:,j}) & \cdots & \sum_{j=1}^{N}\boldsymbol{A}_{j,N}\boldsymbol{h}(\boldsymbol{X}_{:,j}) \\ \boldsymbol{0} & \cdots & \boldsymbol{0} \end{pmatrix}.$$

We define

$$\mathcal{E}_{in}(\boldsymbol{X}) = \begin{pmatrix} \boldsymbol{X} \\ \boldsymbol{0}_{(3(n-1)N-D)^+ \times N} \end{pmatrix},$$

and

$$\mathcal{E}_{out}(\boldsymbol{X}) = \boldsymbol{X}_{1:2,1:N}.$$

In the last, we let $\mathcal{F} := \mathcal{E}_{out} \circ \mathcal{F}_{SA} \circ \mathcal{F}_{FF} \circ \mathcal{E}_{in}$. Through our consctuction above, it is clear that

$$\mathcal{F} \in \mathcal{T}(\max\{3(n-1)N, D\}, 1, 1, 3(n-1)N, 1).$$

The proof is completed by direct verify that

$$\mathcal{F}(\boldsymbol{X}^{(i)})_{:,m_i} \neq \mathcal{F}(\boldsymbol{X}^{(j)})_{:,m_j}$$

for any $i \neq j \in [n]$. $\qquad\square$

## G   PROOF OF SECTION 6

### G.1   PROOF OF THEOREM 6.1

*Proof of Theorem 6.1.*   **Step 1:** Indentify [MASK] token in different contexts. Note that the [MASK] token in all data points are the same, the following Lemma implies that we can find a single self-attention layer to map each [MASK] to different values according to the contexts. By applying Lemma 5.1x to $(\boldsymbol{X}^{(1)}, \boldsymbol{y}^{(1)}), \cdots, (\boldsymbol{X}^{(n)}, \boldsymbol{y}^{(n)})$, we know that there exists an self-attention layer $\boldsymbol{\mathcal{F}}_{SA} \in \mathcal{T}(D, 1, 1, 1)$ which has the following form

$$\boldsymbol{\mathcal{F}}_{SA}(\boldsymbol{X}) = \boldsymbol{X} + \boldsymbol{W}_O \boldsymbol{W}_V \boldsymbol{X} \sigma_S \left[ (\boldsymbol{W}_K \boldsymbol{X})^\top (\boldsymbol{W}_Q \boldsymbol{X}) \right]$$

such that

$$\boldsymbol{\mathcal{F}}_{SA}(\boldsymbol{X}^{(i)})_{:,m_i} \neq \boldsymbol{\mathcal{F}}_{SA}(\boldsymbol{X}^{(j)})_{:,m_j} \quad \text{for any } i \neq j \in [n].$$

Define the embedding layer $\mathcal{E}_{in}$ as

$$\mathcal{E}_{in}(\boldsymbol{X}) := \begin{pmatrix} \boldsymbol{X} & \boldsymbol{0} \\ \boldsymbol{1}_{1 \times N} & \boldsymbol{0} \\ \boldsymbol{0}_{(3n-D)^+ \times N} & \boldsymbol{0} \\ \boldsymbol{0} & 1 \end{pmatrix} \in \mathbb{R}^{(\max\{3n,D\}+2) \times (N+1)}.$$

Then, we define the following matrices to adapt to the embedded input

$$\boldsymbol{W}_K^{(0)} := \begin{pmatrix} \boldsymbol{W}_K & \boldsymbol{0} \end{pmatrix} \in \mathbb{R}^{1 \times (\max\{3n,D\})},$$

$$\boldsymbol{W}_Q^{(0)} := \begin{pmatrix} \boldsymbol{W}_Q & \boldsymbol{0} \end{pmatrix} \in \mathbb{R}^{1 \times (\max\{3n,D\})},$$

$$\boldsymbol{W}_V^{(0)} := \begin{pmatrix} \boldsymbol{W}_V & \boldsymbol{0} \end{pmatrix} \in \mathbb{R}^{1 \times (\max\{3n,D\})},$$

$$\boldsymbol{W}_O^{(0)} := \begin{pmatrix} \boldsymbol{W}_O \\ \boldsymbol{0} \end{pmatrix} \in \mathbb{R}^{(\max\{3n,D\}) \times 1}.$$

Construct self-attention layer $\boldsymbol{\mathcal{F}}_{SA}^{(0)}$ by $\boldsymbol{W}_K^{(0)}, \boldsymbol{W}_Q^{(0)}, \boldsymbol{W}_V^{(0)}, \boldsymbol{W}_O^{(0)}$ and the mask matrix $\boldsymbol{P}_0$, that is

$$\boldsymbol{\mathcal{F}}_{SA}^{(0)}(\mathcal{E}_{in}(\boldsymbol{X})) := \mathcal{E}_{in}(\boldsymbol{X}) + \boldsymbol{W}_O^{(0)} \boldsymbol{W}_V^{(0)} \mathcal{E}_{in}(\boldsymbol{X}) \sigma_S \left[ \left( \boldsymbol{W}_K^{(0)} \mathcal{E}_{in}(\boldsymbol{X}) \right)^\top \left( \boldsymbol{W}_K^{(0)} \mathcal{E}_{in}(\boldsymbol{X}) \right) + \boldsymbol{P}_0 \right]$$

$$= \mathcal{E}_{in}(\boldsymbol{X}) + \begin{pmatrix} \boldsymbol{W}_O \boldsymbol{W}_V \boldsymbol{X} \sigma_S \left[ (\boldsymbol{W}_K \boldsymbol{X})^\top (\boldsymbol{W}_Q \boldsymbol{X}) \right] & \boldsymbol{0} \\ \boldsymbol{0} & \boldsymbol{0} \end{pmatrix}$$

$$= \begin{pmatrix} \boldsymbol{\mathcal{F}}_{SA}(\boldsymbol{X}) & \boldsymbol{0} \\ \boldsymbol{1}_{1 \times N} & \boldsymbol{0} \\ \boldsymbol{0}_{(3n-D)^+ \times N} & \boldsymbol{0} \\ \boldsymbol{0} & 1 \end{pmatrix}.$$

**Step 2:** Point fitting by FFNs.

Recall that the effect of $\boldsymbol{\mathcal{F}}_{SA}$ is to distinguish the [MASK] token in each input, that is

$$\boldsymbol{\mathcal{F}}_{SA}(\boldsymbol{X}^{(i)})_{:,m_i} \neq \boldsymbol{\mathcal{F}}_{SA}(\boldsymbol{X}^{(j)})_{:,m_j}.$$

By applying the Lemma H.4 to the set $\left\{ \boldsymbol{\mathcal{F}}_{SA}(\boldsymbol{X}^{(i)})_{:,m_i} \right\}_{i \in [n]}$, we know that there exists a $\sigma_R$-activated FFN $\boldsymbol{f} \in \mathcal{NN}(3n, 1, \mathbb{R}^D \to \mathbb{R})$ such that

$$\boldsymbol{f}(\boldsymbol{\mathcal{F}}_{SA}(\boldsymbol{X}^{(i)})_{:,m_i}) = y_i \quad \text{for any } i \in [n].$$

Then, by applying Proposition 4.1 to $\boldsymbol{f}$, there exists an attention-only Transformer $\boldsymbol{\mathcal{F}}_1 \in \mathcal{T}(\max\{3n, D\} + 2, \max\{3n, D\} + 1, 1, 1)$ and $M > 0$ with the following form

$$\boldsymbol{\mathcal{F}}_1(\boldsymbol{X}) = \boldsymbol{\mathcal{E}}_{out} \circ \boldsymbol{\mathcal{F}}_{SA}^{(1)} \circ \boldsymbol{\mathcal{E}}_{in}(\boldsymbol{X})$$

satisfying

$$\|\boldsymbol{\mathcal{F}}_1(\boldsymbol{X}) - \boldsymbol{f}(\boldsymbol{X})\|_\infty < \varepsilon \quad \text{for any } \boldsymbol{X} \in [-M, M]^{D \times N}.$$

Define $\boldsymbol{\mathcal{F}}$ as

$$\boldsymbol{\mathcal{F}}(\boldsymbol{X}) := \boldsymbol{\mathcal{E}}_{out} \circ \boldsymbol{\mathcal{F}}_{SA}^{(1)} \circ \boldsymbol{\mathcal{F}}_{SA}^{(0)} \circ \boldsymbol{\mathcal{E}}_{in}(\boldsymbol{X}).$$

The proof is completed by verifying that for any $\varepsilon > 0$

$$\left| \boldsymbol{\mathcal{F}}(\boldsymbol{X}^{(i)})_{:,m_i} - y_i \right| < \varepsilon \quad \text{for any } i \in [n],$$

and

$$\boldsymbol{\mathcal{F}} \in \mathcal{T}(\max\{3n, D\} + 2, \max\{3n, D\} + 1, 1, 2).$$

$\square$

## H  SUPPORTING LEMMAS

**Lemma H.1** (Park et al. (2021)). *Let $d \in \mathbb{N}$. Then, for any finite subst $\mathcal{X} \subset \mathbb{R}^d$, there exists a unit vector $\boldsymbol{v} \in \mathbb{R}^d$ such that*

$$\frac{1}{|\mathcal{X}|^2} \sqrt{\frac{8}{\pi d}} \|\boldsymbol{x} - \boldsymbol{x}'\| \leq \|\boldsymbol{v}'(\boldsymbol{x} - \boldsymbol{x}')\| \leq \|\boldsymbol{x} - \boldsymbol{x}'\| \leq \|\boldsymbol{x} - \boldsymbol{x}'\|.$$

**Lemma H.2** (Kajitsuka & Sato (2023)). *Given a finite subset $\mathcal{X} \subset \mathbb{R}^d$. Then, for any $\delta > 0$, there exists matrices $\boldsymbol{W}_K, \boldsymbol{W}_Q \in \mathbb{R}^{1 \times d}$ such that*

$$\left| (\boldsymbol{W}_K \boldsymbol{x}_a)^\top (\boldsymbol{W}_Q \boldsymbol{x}_c) - (\boldsymbol{W}_K \boldsymbol{x}_b)^\top (\boldsymbol{W}_Q \boldsymbol{x}_c) \right| > \delta$$

*for any $\boldsymbol{x}_a, \boldsymbol{x}_b, \boldsymbol{x}_c \in \mathcal{X}$ with $\boldsymbol{x}_a \neq \boldsymbol{x}_b$.*

*Proof of Lemma H.2.* By applying Lemma H.1 to $\mathcal{X} \cup \{0\}$, we know that there exists a unit vector $\boldsymbol{v} \in \mathbb{R}^d$ such that for any $\boldsymbol{x}_a, \boldsymbol{x}_b \in \mathcal{X} \cup \{0\}$ such that $\boldsymbol{x}_a \neq \boldsymbol{x}_b$, we have

$$\frac{1}{(|\mathcal{V} + 1|)^2} \sqrt{\frac{8}{\pi d}} \|\boldsymbol{x}_a - \boldsymbol{x}_b\| \leq |\boldsymbol{v}^\top (\boldsymbol{x}_a - \boldsymbol{x}_b)| \leq \|\boldsymbol{x}_a - \boldsymbol{x}_b\|$$

In particular, if we let one of $\boldsymbol{x}_a$ and $\boldsymbol{x}_b$ be 0, we have

$$\frac{1}{(|\mathcal{V} + 1|)^2} \sqrt{\frac{8}{\pi d}} \|\boldsymbol{x}_c\| \leq |\boldsymbol{v}^\top \boldsymbol{x}_c| \leq \|\boldsymbol{x}_c\| \quad \text{for any } \boldsymbol{x}_c \in \mathcal{X}.$$

Let $\varepsilon = \min\{\|\boldsymbol{x}_a - \boldsymbol{x}_b\| \mid \boldsymbol{x}_a \neq \boldsymbol{x}_b \in \mathcal{X}\}$ and $r = \min\{\|\boldsymbol{x}\| \mid \boldsymbol{x} \in \mathcal{X}\}$. Assume that $r \neq 0$. Then, we can pick up arbitrary vectors $\boldsymbol{u}, \boldsymbol{u}' \in \mathbb{R}$ with

$$|\boldsymbol{u}\boldsymbol{u}'| = (|\mathcal{X}| + 1)^4 \frac{\pi d}{8} \frac{\delta}{\varepsilon r}.$$

Let $\boldsymbol{W}_K = \boldsymbol{u}\boldsymbol{v}^\top$ and $\boldsymbol{W}_Q = \boldsymbol{u}'\boldsymbol{v}^\top$, we have

$$\left| (\boldsymbol{W}_K \boldsymbol{x}_a)^\top (\boldsymbol{W}_Q \boldsymbol{x}_c) - (\boldsymbol{W}_K \boldsymbol{x}_b)^\top (\boldsymbol{W}_Q \boldsymbol{x}_c) \right|$$

$$= \left| (\boldsymbol{x}_a - \boldsymbol{x}_b)^\top (\boldsymbol{W}_K)^\top (\boldsymbol{W}_Q \boldsymbol{x}_c) \right|$$

$$= \left| (\boldsymbol{x}_a - \boldsymbol{x}_b)^\top \boldsymbol{v} \right| \cdot |\boldsymbol{u}\boldsymbol{u}'| \cdot \left| \boldsymbol{v}^\top \boldsymbol{x}_c \right|$$

$$\geq \delta,$$

which completes the proof. $\square$

**Lemma H.3** (Kajitsuka & Sato (2023))**.** *Let $\boldsymbol{a}^{(1)}, \cdots, \boldsymbol{a}^{(n)} \in \mathbb{R}^d$ be pair-wise distinct vectors satisfying*

$$\|\boldsymbol{a}^{(i)} - \boldsymbol{a}^{(j)}\| > 2\log n + 3 \quad \text{for any } i \neq j \in [n].$$

*Then, the following holds*

$$\left(\boldsymbol{a}^{(i)}\right)^\top \sigma_S \left[\boldsymbol{a}^{(i)}\right] \neq \left(\boldsymbol{a}^{(j)}\right)^\top \sigma_S \left[\boldsymbol{a}^{(j)}\right]$$

*for any $i \neq j \in [n]$.*

**Lemma H.4** (Point fitting by ReLU FFNs, modified from (Jiao et al., 2025a))**.** *Let $(\boldsymbol{x}_1, y_1), \cdots, (\boldsymbol{x}_n, y_n)$ be a set of input-output pairs such that $\boldsymbol{x}_i \in \mathbb{R}^d$, $y_i \in \mathbb{R}$ and $\boldsymbol{x}_i \neq \boldsymbol{x}_j$ if $i \neq j$. Then, there exists a $\sigma_R$-activated feed-forward network $\boldsymbol{f} \in \mathcal{NN}(3n, 1, \mathbb{R}^d \to \mathbb{R})$ such that*

$$\boldsymbol{f}(\boldsymbol{x}_i) = y_i \quad \text{for any } i \in [n].$$

*Proof of Lemma H.4.* Let $K > 0$ be determined later. Since $\boldsymbol{x}_i$ are pair-wise distnct. According to Lemma H.1, there exists a vector $\boldsymbol{v} \in \mathbb{R}^d$ such that $\boldsymbol{v}^\top \boldsymbol{x}_i, i \in [n]$ are also pair-wise distinct. We define

$$\boldsymbol{W}_1^{(i)} = K \begin{pmatrix} 1 \\ 1 \\ 1 \end{pmatrix} \boldsymbol{v}^\top, \quad \boldsymbol{b}_1^{(i)} = \begin{pmatrix} -K\boldsymbol{v}^\top \boldsymbol{x}_i - 1 \\ -K\boldsymbol{v}^\top \boldsymbol{x}_i \\ -K\boldsymbol{v}^\top \boldsymbol{x}_i + 1 \end{pmatrix}, \quad \boldsymbol{W}_2^{(i)} = y_i \begin{pmatrix} 1 & -2 & 1 \end{pmatrix}, \quad \boldsymbol{b}_2^{(i)} = \boldsymbol{0},$$

where $\boldsymbol{W}_1^{(i)} \in \mathbb{R}^{3 \times d}$, $\boldsymbol{b}_1^{(i)} \in \mathbb{R}^3$, $\boldsymbol{W}_2^{(i)} \in \mathbb{R}^{1 \times 3}$, $\boldsymbol{b}_2^{(i)} \in \mathbb{R}$. It is straightforward to verify that the following holds for any $\boldsymbol{x} \in \mathbb{R}^d$

$$\boldsymbol{W}_2^{(i)} \sigma_R \left(\boldsymbol{W}_1^{(i)} \boldsymbol{x} + \boldsymbol{b}_1^{(i)}\right) + \boldsymbol{b}_2^{(i)}$$

$$= y_i \left(\sigma_R \left(K\boldsymbol{v}^\top (\boldsymbol{x} - \boldsymbol{x}_i) - 1\right) - 2\sigma_R \left(K\boldsymbol{v}^\top (\boldsymbol{x} - \boldsymbol{x}_i)\right) + \sigma_R \left(K\boldsymbol{v}^\top (\boldsymbol{x} - \boldsymbol{x}_i) + 1\right)\right)$$

$$= y_i \boldsymbol{I}_i(\boldsymbol{x}),$$

where $\boldsymbol{I}_i(\boldsymbol{x})$ satisfies $\boldsymbol{I}_i(\boldsymbol{x}_i) = 1$ and $\boldsymbol{I}_i(\boldsymbol{x}_i) = 0$ if $\left|\boldsymbol{v}^\top (\boldsymbol{x} - \boldsymbol{x}_i)\right| \geq \frac{1}{K}$. So, we choose

$$K > \frac{2}{\min_{i \neq j} \left|\boldsymbol{v}^\top (\boldsymbol{x}_i - \boldsymbol{x}_j)\right|}.$$

Define

$$\boldsymbol{W}_1 = \begin{pmatrix} \boldsymbol{W}_1^{(1)} \\ \vdots \\ \boldsymbol{W}_1^{(n)} \end{pmatrix}, \quad \boldsymbol{b}_1 = \begin{pmatrix} \boldsymbol{b}_1^{(1)} \\ \vdots \\ \boldsymbol{b}_1^{(n)} \end{pmatrix}, \quad \boldsymbol{W}_2 = \begin{pmatrix} \boldsymbol{W}_2^{(1)} & \cdots & \boldsymbol{W}_2^{(n)} \end{pmatrix}, \quad \boldsymbol{b}_2 = \boldsymbol{0}$$

where $\boldsymbol{W}_1 \in \mathbb{R}^{3n \times d}$, $\boldsymbol{b}_1 \in \mathbb{R}^{3d}$, $\boldsymbol{W}_2 \in \mathbb{R}^{1 \times 3n}$, $\boldsymbol{b}_2 \in \mathbb{R}$. Let

$$\boldsymbol{f}(\boldsymbol{x}) = \boldsymbol{W}_2 \sigma_R \left(\boldsymbol{W}_1 \boldsymbol{x} + \boldsymbol{b}_1\right) + \boldsymbol{b}_2$$

$$= \sum_{i=1}^n y_i \boldsymbol{I}_i(\boldsymbol{x}).$$

It is clear that $\boldsymbol{f} \in \mathcal{NN}(3n, 1, \mathbb{R}^d \to \mathbb{R})$. The proof is completed by pointing out that

$$\boldsymbol{f}(\boldsymbol{x}_i) = y_i \quad \text{for any } i \in [n].$$

$\square$

# I  THE USE OF LARGE LANGUAGE MODELS

The LLMs are used in the following aspects:

- Improving the clarity of writing and grammar.
- Drafting code snippets for preliminary experiments.
- Summarizing related literature for internal discussion.

We carefully reviewed and revised any text or code suggested by LLMs, and remain fully responsible for the scientific accuracy, originality, and integrity of this work.

