# OpenReview forum: "Exploring Non-linearity in Attention"
_ICLR.cc/2026/Conference — ICLR 2026 Conference Withdrawn Submission_

### Official Review · Reviewer_WrGj · 2025-10-20

**Soundness:** 4
**Presentation:** 3
**Contribution:** 2
**Rating:** 2
**Confidence:** 3

**Summary:**

The authors study two theoretical questions related to (non-causal) attention in transformers within the setting of masked language modeling: (1) whether a series of stacked attention layers (with a minor modification, namely, adding a bias vector) can approximate an MLP applied to each token in the input sequence in parallel; (2) whether fixing the attention matrix or randomizing it independently of the input can result in simple models (just an attention layer or an attention + MLP respectively) can distinguish the mask tokens for different inputs by assigning the corresponding output tokens different values. As an application of these results, the authors show that a network built from two attention layers, with the first layer using a construction of (2) which enables the attention layer to distinguish mask tokens of different inputs, and the second layer using the bias vector to approximate an MLP as in (1), is sufficient to solve a masked language modeling problem to arbitrarily high accuracy, using a model dimension and number of attention heads which are both linear in the token dimension and number of training samples (and with the per-head dimension being 1). Simple synthetic experiments are included to provide intuition about claims (1) and (2).

**Strengths:**

- The problem settings are novel: while universal approximation-type results have been obtained for general equivariant sequence-to-sequence functions, approximating an MLP applied to each token has not been considered, to the best of my knowledge. Similarly studying the approximation power of an attention layer with input-independent attention is novel to the best of my knowledge.
- Because the setting is so restricted, the rates are reasonably sharp, requiring model size which is linear in the number of samples and the token dimension (cf universal approximation rates which often can require exponentially large models).
- The continued application to (a simplified version of) masked language modeling helps keep the paper relatively concrete, increasing its accessibility and making it easier to read.

**Weaknesses:**

__Major__
- While the setting is novel, it is not particularly relevant. Using solely attention layers to approximate an MLP is only a meaningful statement about the approximation result of neural networks if these neural networks are entirely composed of iterated attention layers (as done in the last Theorem in the main body). Such architectures are not often used. Moreover, input-independent attention maps are impractical and basically defeat the point of attention operators.
- Moreover, the constructions used in the existence proofs also reduce the potential impact --- they require a per-head dimension of 1 and a number of attention heads which is either 1 or linear in the number of samples and the token dimension. None of these scalings are used in practice due to poor empirical performance and efficiency.
- The definition of context-dependence used in Section 5 seems altogether too loose to be interesting (see below question Q1), namely it might be easy to show that these non-equalities hold with overwhelming probability for operators that are not self-attention based, or with weights chosen randomly without a specific construction (with overwhelming probability, say). As written, I don't think I understand the significance or importance of any result in Section 5.

__Minor__
- Definition 3.1: "Mased Language Models" -> "Masked Language Models"
- Remark 4.2: I would really disagree with your interpretation of fixing some parameters and training the rest as equivalent to prompt tuning. You point out the essential difference in the remark: prompt tuning does not require tuning any parameters but rather just changing the inputs! Keeping some parameters fixed does not make your construction similar to prompt tuning; by this logic you can also argue that last-layer finetuning of a language model is in some sense "prompt tuning", but this would not be appropriate at all.

**Questions:**

Q1: Is there an example of a non-trivial sequence-to-sequence operator which does not satisfy the distinguishing-different-contexts property shown by the conclusions of Propositions 5.1, 5.2, 5.3? The formal definition of this property is not exactly included in your paper but you can use [1] Definition 3.1 (which is cited in your paper) for a reference.

Q2: What are the main technical issues preventing the use of more practical configurations and (input-dependent) softmax attention for proving a result similar to Theorem 6.1? Do they pose prohibitive difficulties? If so, how does this work significantly progress beyond [2], which you briefly cited in this work (is it just that you can get sharper rates by working within a more specific context)?

[1]: Yun, Chulhee, et al. "Are transformers universal approximators of sequence-to-sequence functions?." arXiv preprint arXiv:1912.10077 (2019).
[2]: Hu, Jerry Yao-Chieh, et al. "Universal Approximation with Softmax Attention." arXiv preprint arXiv:2504.15956 (2025).

---

> ### Author Response · Authors · 2025-11-21
>
> We sincerely thank the reviewer for their constructive and insightful comments, which will undoubtedly help improve this paper. We recognize that the novelty and significance of our work are relatively limited, and we have therefore decided to withdraw the submission. Before doing so, however, we would like to respond to several of the reviewer’s questions—not as a last attempt to salvage the paper, but simply to share our thoughts.
>
> > **Q1: Is there an example of a non-trivial sequence-to-sequence operator which does not satisfy the distinguishing-different-contexts property shown by the conclusions of Propositions 5.1, 5.2,**
>
> You may refer to Theorem 1 in [1]. One self-attention layer with hardmax activation function does not satisfy this property. Similarly, one self-attention layer with ReLU activation function does not satisfy this property either [2].
>
> > "**Q2: What are the main technical issues preventing the use of more practical configurations and (input-dependent) softmax attention for proving a result similar to Theorem 6.1? Do they pose prohibitive difficulties? If so, how does this work significantly progress beyond [2], which you briefly cited in this work (is it just that you can get sharper rates by working within a more specific context)?**"
>
> Actually, the self-attention used in Theorem 6.1 is input-dependent softmax self-attention. You may refer to the discussion under the statement of Theorem 6.1. We can extend this result to an input-independent self-attention mechanism based on Proposition 5.2 and 5.3. I think the biggest difference between our work and [3] is that the way attention heads approximate FFN is totally different. Thanks for your comments, which lead us to rethink the relationship between our work and [3].
>
> > **Weakness: Remark 4.2: I would really disagree with your interpretation of fixing some parameters and training the rest as equivalent to prompt tuning. You point out the essential difference in the remark: prompt tuning does not require tuning any parameters but rather just changing the inputs! Keeping some parameters fixed does not make your construction similar to prompt tuning; by this logic you can also argue that last-layer finetuning of a language model is in some sense "prompt tuning", but this would not be appropriate at all.**
>
> We apologize for the use of the word “similar” in our original paper and we did not anticipate that it would lead to such a significant misunderstanding. We have never considered these two mechanisms to be equivalent. What we intended to convey is that adding a fixed bias vector to the inputs shares a conceptual resemblance to prompt tuning but also with some difference. In our theoretical construction, this fixed vector is untrained and can be chosen randomly under mild conditions, whereas in prompt tuning, the appended vectors (i.e., prompts) are trainable while the backbone model is kept frozen.
>
> The purpose of Remark 4.2 was to draw a connection between our construction and prompt tuning, and to raise the question of whether a similar idea: making the fixed vector trainable and longer, could further help improve performance. In fact, this idea was empirically explored much earlier in [4].
>
> **Reference**
>
> [1] Are Transformers with One Layer Self-Attention Using Low-Rank Weight Matrices Universal Approximators?
>
> [2] Prompt Tuning Transformers for Data Memorization.
>
> [3] Universal Approximation with Softmax Attention.
>
> [4] Augmenting self-attention with persistent memory.

---

> ### Comment · Reviewer_WrGj · 2025-11-22
> **Reply to Authors**
>
> Thanks for the clarifications. For posterity, I will add some concluding feedback given the authors' responses.
>
> > Reply to Q1
>
> Thanks; I think future iterations of the paper should make this remark explicitly; even though the paper sources the ideas of the definition, the paper may need to establish that the definition is a non-trivial property. When I first saw it I thought it was trivial.
>
> > Reply to Q2
>
> Thanks for the clarification. I believe the theorem statement is not exactly clear that the constructed network has an instance-dependent softmax, and so in future iterations of the work I recommend the paper clarifies this. Note that at the bottom of Page 7, the paper reads: "we allow Transformers to include both standard self-attention layers
> and fixed-pattern self-attention layers". Theorem 6.1 does not clarify which type of attention is used. But the discussion below the theorem states that the proof uses a construction from Section 5, which is about instance-independent softmax operators (compare the first paragraph of Section 5 with the paragraph after Theorem 6.1).
>
> Since one of the main questions and one of the main weaknesses were clarified to have already been addressed (modulo some writing to clear up), I will raise my score. But the more significant concerns about the formulation and constructions remain.
>
> If you continue to withdraw, best wishes with future iterations of this paper.

---

> > ### Author Response · Authors · 2025-11-28
> >
> > Thank you very much for raising the score and your best wishes!

---

### Official Review · Reviewer_xko5 · 2025-10-23

**Soundness:** 3
**Presentation:** 3
**Contribution:** 2
**Rating:** 6
**Confidence:** 3

**Summary:**

This paper investigates the role of non-linearity across different components of the Transformer model. The authors first proof that the FFN  layer can be replaced by a specialized Self-Attention layer that utilizes additional registers and specific mask patterns. Next,  motivated by attention patterns observed in BERT, the paper suggests and proves that fixed linear attention (trainable or not) is capable of performing the Masked Language Modeling (MLM) task, given the inclusion of an additional FFN layer. This is further validated using an artificial MLM task, where the authors show that a model with two layers of attention is sufficient for the task. The authors also includes  an experimental appendix that shows the theory in practice (in a controlled setting).

**Strengths:**

S1: The idea of investigating the non-linearity in transformer models is a valuable and interesting research direction.

S2: The second part of the paper regarding contextual non-linearity is non-trivial and the results are surprising, even under the simplified assumptions used (which are simpler than actual native text assumptions).

S3: The inclusion of motivation from pre-trained LLMs like BERT is an important aspect of this work.

S4: Appendix D containing the experiments is important and validates the practical implications of the paper. It is suggested that this material should be moved into the main paper (at least partially).

**Weaknesses:**

W1: The first part of the paper regarding position-wise non-linearity is somewhat trivial. Although a formal proof for replacing the FFN with Self-Attention may be novel, the concept that the FFN acts as Self-Attention on each individual token independently (can be done with a diagonal attention pattern) is simple.

W2: Given that most of the paper's proofs are in the appendices, many of the preliminaries could also be moved to the appendix to significantly improve the readability of the main body of the paper.

W3: The observation regarding BERT attention heads is widely known and has been demonstrated in several papers ([1, 2]). While this is acknowledged as not being the main contribution, the authors should consider using more modern encoder models (e.g., those from the MTEB leaderboard [3]) to draw evidence for these patterns.

W4: Minor: The paper should cite existing work, such as [2, 4], at least as motivation, and clearly explain how those findings relate to the current paper.


[1] https://arxiv.org/abs/1906.04341

[2] https://arxiv.org/abs/2211.03495

[3] https://huggingface.co/spaces/mteb/leaderboard

[4] https://arxiv.org/abs/2412.11965

**Questions:**

Q1: I think there is one minor typo is in line 264: "out" instead of "our".

---

> ### Author Response · Authors · 2025-11-21
>
> We are extremely grateful for your comments and for your recognition of our contributions. Your positive evaluation was very encouraging at a time when we were feeling quite discouraged. However, after considering the feedback from the other reviewers, we have decided to withdraw this submission. We believe that, with your very helpful suggestions, we will be able to significantly improve this paper.

---

> > ### Comment · Reviewer_xko5 · 2025-11-25
> >
> > Thanks for your response. Good Luck with the  re-submission.

---

### Official Review · Reviewer_fbPb · 2025-10-28

**Soundness:** 2
**Presentation:** 2
**Contribution:** 2
**Rating:** 2
**Confidence:** 4

**Summary:**

This paper primarily investigates two sources of non-linearity in Transformer architectures:
- Position-wise Non-linearity: Provided by feed-forward layers, applying non-linear transformations independently to each token
- Contextual Non-linearity: Provided by self-attention mechanisms, generating input-dependent attention patterns through dot products and Softmax

It proves that stacked self-attention layers can approximate deep feed-forward networks by appending a fixed bias token and using masks to limit interactions. Analyses of pre-trained BERT show recurring heads that mainly interact with [CLS]/[SEP] and behave like FFNs, and synthetic experiments validate that attention with a fixed vector approximates FFNs best when the number of layers matches the FFN depth, with performance improving as heads increase. The paper then shows contextual non-linearity is not indispensable. It then constructs a two-layer attention-only Transformer for masked language modeling. Empirically, repeated attention patterns, context-driven divergence of [MASK] embeddings with depth, small value norms for special tokens, and strong performance with fixed or random patterns on synthetic MLM support the theory.

**Strengths:**

There are mainly two theoretical contributions:
- Contribution 1 (Theorem 4.1): Proves that by appending a fixed bias vector to the input, stacked self-attention layers can approximate deep feed-forward networks. Demonstrates that attention mechanisms alone are sufficient to implement position-wise non-linearity.
- Contribution 2 (Propositions 5.2 and 5.3): Proves that contextual non-linearity is not indispensable. Fixed or even randomly chosen attention patterns, combined with feed-forward layers, can still produce context-sensitive representations.

And for applications and experimental validations:

- Application (Theorem 6.1): Proves that a two-layer attention-only Transformer can accurately predict masked tokens in masked language modeling tasks
- Experimental Validation: Observes that certain attention heads in pre-trained BERT models exhibit feed-forward-like behavior
Validates theoretical results on synthetic data

**Weaknesses:**

My biggest concern is that the paper’s central questions about where non-linearity comes from in Transformers are already well understood, both theoretically and empirically. The contribution in this paper seems to be incremental and framed under contrived constructions.

- On the attention side, recent work formalizes how Transformers implement functional gradient descent and learn non-linear functions from context in creating non-linear mappings[1][2][5].

- On the MLP/channel-mixing side, the choice and design of activation functions has long focused on non-linearity and expressivity, and GLU-style variants have repeatedly demonstrated that enriching the feed-forward non-linearity yields consistent empirical gains and practical benefits [3][7].

- Foundational analyses within BERT also dissect where non-linearity lives and how it interacts with commutativity, adding to a body of evidence that the key non-linear components and their effects are already known and measurable in standard architectures and training regimes [4].

- There is also contradictory conclusion from [6] which empirically proves the near-linearity of Transformers.

Also, some conclusions in this paper is obviously not practical:

- As mentioned in Theorem 4.1 w/ attention-only architecture, I don't believe this makes sense considering MLP contains knowledge in language models [8].

- Propositions 5.2 and 5.3, fixed or even randomly chosen attention patterns CAN produce context-sensitive representations because it still a metaformer architecture [9] i.e., w/ time-mixing. This is already a well-understood conclusion. But this is not a good time mixing module for training a good Transformer.

[1] Transformers Implement Functional Gradient Descent to Learn Non-Linear Functions In Context, https://www.arxiv.org/abs/2312.06528v6

[2] How Do Nonlinear Transformers Learn and Generalize in In-Context Learning?, https://www.arxiv.org/abs/2402.15607

[3] GLU Attention Improve Transformer, https://www.arxiv.org/abs/2507.00022

[4] Of Non-Linearity and Commutativity in BERT, https://www.arxiv.org/abs/2101.04547

[5] Transformers learn in-context by gradient descent, https://www.arxiv.org/abs/2212.07677

[6] Your Transformer is Secretly Linear, https://www.arxiv.org/abs/2405.12250

[7] GLU Variants Improve Transformer, https://www.arxiv.org/abs/2002.05202

[8] Neuron-Level Knowledge Attribution in Large Language Models https://www.arxiv.org/abs/2312.12141

[9] MetaFormer: A Unified Meta Framework for Fine-Grained Recognition https://www.arxiv.org/abs/2203.02751

**Questions:**

Please refer to the weakness part.

---

> ### Author Response · Authors · 2025-11-21
>
> We sincerely thank the reviewer for these constructive and insightful comments, from which we have learned a great deal. We understand—and have been aware throughout the writing process—that the conclusions presented in this article have already been well observed in previous empirical studies, a point we repeatedly emphasize in the paper. Our contribution was to provide theoretical explanations for these phenomena. However, we recognize that the new insights this work offers to the community are quite limited, and this has led us to the decision to withdraw the submission.

---

> ### Comment · Reviewer_fbPb · 2025-11-21
>
> Thanks for the reply. Good Luck for the submission in the future! : ).

---

### Official Review · Reviewer_GnFK · 2025-10-31

**Soundness:** 3
**Presentation:** 2
**Contribution:** 1
**Rating:** 2
**Confidence:** 3

**Summary:**

In this paper, the authors study the role of non-linearity in transformer networks, from a theoretical perspective. The authors note that there are two sources of non-linearity in transformers: the self-attention and the feed-forward modules. The authors argue that these two potentially play different roles, as the non-linearity in self-attention tend to capture interactions between tokens of the sequence, while the non-linearity in feed-forward is applied independently to each token (referred to as "position-wise" non-linearity). This leads to two questions that the authors aim to answer from a theoretical perspective:
- Is position-wise non-linearity only captured in feed-forward modules?
- Is contextual non-linearity necessary for transformer networks?

To answer the first question, the authors show that the self-attention mechanism can approximate feed-forward networks with a single hidden layer, hence replacing feed-forward modules. To answer the second question, the authors explore the idea of replacing context dependent attention patterns by fixed attention patterns (which do not depend on the input). They show that under mild conditions on the input, for any fixed attention pattern, there exists a feed-forward network such that the composition of the feed-forward and the fixed attention mechanism can "separate" the n input examples, hence allowing to learn the mapping from these n input example to the corresponding outputs. Finally, the authors perform experiments to validate their theoretical results. In particular, they show that in BERT, there are heads exhibiting the behavior described in paper, namely that some heads compute something similar to a feed-forward networks (by interacting mostly with the [cls] token), while some other heads have a fixed pattern, for example only looking at the preceding token.

**Strengths:**

The paper is relatively well written and easy to follow.

**Weaknesses:**

My main concerns with this paper are about novelty and significance of the results. The paper study the two following questions (1) could self-attention replace feed-forward modules in transformers and (2) can fixed attention patterns replace self-attention in transfomers. To me, these two questions are orthogonal, since both modifications could not be applied simultaneously. Indeed, the results corresponding to answer question (1) require the attention pattern to depend on the input.

The first question was already explored in previous work, both theoretically and empirically. In particular, Huben & Morris already proved that the self attention pattern with one head can implement a MLP neuron using generalized SiLU non-linearity, hence allowing to also approximate ReLU and GeLU networks. I believe that the constructions and proofs in this paper are similar to the one from Huben & Morris.

Second, the results regarding fixed attention patterns do not seem very significant either. In particular, I believe that similar results could be obtained by applying the universal approximation theorem. In particular, since the result apply to any fixed attention pattern, this means that the result is also true for the "average operation". To me, this means that the result is not specific to attention, and thus does not bring a lot of understanding of what these networks can and cannot learn.

Finally, the experimental results showing that BERT models exhibit patterns that are described in this work were also already explored in previous work.

Overall, I do not believe that this paper brings much to the understanding of what transformer networks can and cannot learn well. The results about replacing MLP by self-attention were already explored in previous work, and the construction considered in this work would lead to self-attention modules with very large number of heads (one head per hidden dimension of the MLP). On the other hand, to me, the results about fixed attention patterns do not bring more understanding compared to the universal approximation theorem.

**Questions:**

How does Theorem 4.1 extend results from Huben & Morris? I think a better discussion of previous results should be included in the paper.

Why focus on masked language modeling? How are the results specific to this setting versus more general problems?

---

> ### Author Response · Authors · 2025-11-21
>
> We sincerely thank the reviewer for their constructive and insightful comments, which will undoubtedly help improve this paper. We recognize that the novelty and significance of our work are relatively limited, and we have therefore decided to withdraw the submission. Before doing so, however, we would like to respond to several of the reviewer’s questions—not as a last attempt to salvage the paper, but simply to share our thoughts.
>
> > **Q1: How does Theorem 4.1 extend results from Huben & Morris? I think a better discussion of previous results should be included in the paper.**
>
> In Huben & Morris, a single self-attention layer is used to replace feed-forward networks (FFNs) with one hidden layer. We extend their results by showing that the composition of multiple self-attention layers can approximate deep feed-forward neural networks, which requires more delicate error control and a more careful configuration of the weights. We fully agree with the reviewer’s suggestion to include a more detailed discussion of our theoretical contribution.
>
> > **Q2: Why focus on masked language modeling? How are the results specific to this setting versus more general problems?**
>
> The reason we focus on masked language modeling is straightforward. Because BERT is pretrained using this objective and relies on special tokens such as [CLS] and [SEP], it is more natural in this setting to introduce the appended fixed-bias vector used in our theoretical formulation. This choice also keeps our experimental setup close to realistic scenarios (see Figures 1 and 2). In fact, our proof techniques and results can be extended seamlessly to more general universal approximation frameworks. As reflected in our references, our work draws substantial inspiration and theoretical tools from that literature.

---

### Note · Authors · 2025-11-28

**Comment:**

We sincerely thank all the reviewers for their time and thoughtful feedback. Their objective, insightful, and constructive comments were extremely helpful to us. After careful consideration, we believe that the current contribution of this paper is not yet strong enough for publication at ICLR. Therefore, we have decided to withdraw our submission.

**Withdrawal Confirmation:**

I have read and agree with the venue's withdrawal policy on behalf of myself and my co-authors.